# Histone H2A Lys130 acetylation epigenetically regulates androgen production in prostate cancer

Thanh Nguyen[1,2,10,11], Dhivya Sridaran [1,2,11], Surbhi Chouhan[1,2],
Cody Weimholt[3,4], Audrey Wilson[1,2], Jingqin Luo [5], Tiandao Li [6],
John Koomen [7], Bin Fang [7], Nagireddy Putluri[8], Arun Sreekumar[8],
Felix Y. Feng [9], Kiran Mahajan[1,2] & Nupam P. Mahajan [1,2,3] ✉

The testicular androgen biosynthesis is well understood, however, how cancer cells gauge dwindling androgen to dexterously initiate its de novo synthesis remained elusive. We uncover dual-phosphorylated form of sterol regulatory element-binding protein 1 (SREBF1), pY673/951-SREBF1 that acts as an androgen sensor, and dissociates from androgen receptor (AR) in androgen deficient environment, followed by nuclear translocation. SREBF1 recruits KAT2A/ GCN5 to deposit epigenetic marks, histone H2A Lys130-acetylation (H2A-K130ac) in *SREBF1*, reigniting de novo lipogenesis & steroidogenesis. Androgen prevents SREBF1 nuclear translocation, promoting T cell exhaustion. Nuclear SREBF1 and H2A-K130ac levels are significantly increased and directly correlated with late-stage prostate cancer, reversal of which sensitizes castration-resistant prostate cancer (CRPC) to androgen synthesis inhibitor, Abiraterone. Further, we identify a distinct CRPC lipid signature resembling lipid profile of prostate cancer in African American (AA) men. Overall, pY-SREBF1/H2A-K130ac signaling explains cancer sex bias and reveal synchronous inhibition of KAT2A and Tyr-kinases as an effective therapeutic strategy.

Steroidogenesis is the multistep process for biosynthesis of steroid hormones, including androgen from cholesterol that influences both normal physiology as well as disease pathology. The androgen biosynthesis from cholesterol in Leydig cells of testis is under the pulsatile control of pituitary luteinizing hormone. This steroidogenesis acquires a significant relevance in prostate cancer; since prostate do not make testosterone, the prostate cancers become dependent on the testicular androgen for its survival. The intra-tumoral steroidogenesis leading to androgen or testosterone synthesis[1], remain unchanged in prostate cancer patients before and after androgen-deprivation therapy[2–5]. CYP17A is the key enzyme for the androgen synthesis from cholesterol[6], its blocking by the inhibitor, Abiraterone (Zytiga), has emerged to be a major therapeutic option for castration-resistant prostate cancer or CRPC[7,8]. Although effective initially, the resistance

[1]Department of Surgery, Cancer Research Building, Washington University in St Louis, 660 Euclid Ave., St Louis, MO 63110, USA. [2]Department of Urology, Cancer Research Building, Washington University in St Louis, 660 Euclid Ave., St Louis, MO 63110, USA. [3]Siteman Cancer Center, Cancer Research Building, Washington University in St Louis, 660 Euclid Ave., St Louis, MO 63110, USA. [4]Department of Pathology & Immunology, Cancer Research Building, Washington University in St Louis, 660 Euclid Ave., St Louis, MO 63110, USA. [5]Division of Public Health Sciences, Cancer Research Building, Washington University in St Louis, 660 Euclid Ave., St Louis, MO 63110, USA. [6]Bioinformatics Research Core, Center of Regenerative Medicine, Department of Developmental Biology, Washington University at St. Louis, St Louis, MO 63110, USA. [7]Molecular Oncology and Molecular Medicine, Moffitt Cancer Center, Tampa, FL 33612, USA. [8]Dan L Duncan Comprehensive Cancer Center, Baylor College of Medicine, Houston, TX 77030, USA. [9]Helen Diller Family Comprehensive Cancer Center, University of California, San Francisco, CA 94158, USA. [10]Present address: Section of Gastroenterology & Hepatology, Department of Medicine, Baylor College of Medicine, Houston, TX 77030, USA. [11]These authors contributed equally: Thanh Nguyen, Dhivya Sridaran. ✉e-mail: nupam@wustl.edu

typically develops in 9–15 months, and even its combination with Enzalutamide, an AR-antagonist, did not last beyond 19.3 months[9]. Interestingly, recurrent CRPCs exhibit increased synthesis of testosterone and dihydrotestosterone (DHT) from cholesterol[10,11], indicating that CRPCs have resourcefully circumvented dependence on testicular androgen and the de novo androgen biosynthesis in the cancer cells is the major cause for the Abiraterone-resistance. Despite being a pivotal issue[12], the precise molecular signaling that ensures incessant supply of androgen in the presence of abiraterone is not elucidated.

Sterol regulatory-element binding proteins (SREBPs) are the master transcription factors that regulate the genes involved in cholesterol biosynthesis and the LDL receptor (LDLR) pathway and thus have emerged to be the crucial regulators of cellular lipid metabolism and homeostasis[13,14]. In the endoplasmic reticulum (ER), SREBPs form heterodimeric complexes with SREBP cleavage activating protein (SCAP) and when levels of cholesterol decrease, promotes SREBF1 exit from ER to Golgi apparatus, where SREBPs are subjected to two-step proteolytic processing to generate N-terminal-cleaved bHLH-LZ transcription factor[15]. It enters the nucleus and stimulate the transcription of cholesterologenic and lipogenic target genes, thus building cholesterol levels in response to its deficiency[16]. The mammalian genome encodes three SREBP isoforms, encoded by two genes, designated SREBF-1a and SREBF-1c (derived from a single gene, *SREBF1*), and SREBF-2[15]. SREBF-1a is a potent activator of all SREBF-responsive genes, including those that mediate the synthesis of cholesterol, fatty acids, phospholipids and triglycerides. Increased SREBF1 expression upon androgen deprivation is implicated in the pathogenesis of prostate[17], and SREBF1-dependent lipogenic program is hyperactivated in prostate tumors[18]. However precisely how cancer cells kick-start de novo androgen synthesis to overcome the effect of androgen synthesis inhibitors remains to be determined. Overall, the relative short-term efficacy of abiraterone reveals two major caveats; first, how cancer cells sense androgen deficiency, and the second, how CRPCs initiates robust de novo lipid biosynthesis, when there is no apparent decrease in cholesterol levels.

Here, we uncover dual phosphorylated-SREBF1 as a sensor of androgen deficiency leading to its nuclear translocation and deposition of H2A-K130ac epigenetic marks to activate a distinct transcription program that includes *SREBF1*. A significant outcome of the pY-SREBF1/H2A-ac130 signaling nexus is rejuvenation of intratumoral cholesterol & androgen biosynthesis, allowing CRPCs to escape the low androgen situation created by Abiraterone. Thus, reversal SREBF1 Tyr673/951-phosphorylation or H2A-K130ac epigenetic mark opens a distinct opportunity to re-sensitize abiraterone-resistant CRPC tumors.

## Results

### Cancer cells accumulate epigenetic marks, H2A-K130ac in response to androgen deprivation

We reasoned that CRPCs must sense the diminishing androgen levels to rebuild its levels when subjected to abiraterone. To identify the mechanism, first, we explored the suitable cell lines; since C4-2B and VCaP cells have ability to form CRPC tumors[19–22], these cells were treated with increasing concentration of abiratetone and surviving cells were determined post-treatment. Both the cell lines exhibited a significant abiraterone resistance with $IC_{50}$ of 6.3 and 6.7 μM, respectively (Supplementary Fig. S1a), suggesting that CRPCs have inherent ability to acquire abiraterone resistance. To explore the potential signaling involved, C4-2B cells were treated with three distinct classes of inhibitors: CPTH2 (an inhibitor of histone lysine acetyltransferase KAT2A), Ponatinib (targets intracellular BCR-ABL kinase) and Linsitinib (insulin receptor or IR, and insulin-like growth factor 1 receptor tyrosine kinase inhibitor). C4-2B cells exhibited most sensitivity for CPTH2 ($IC_{50}$ 0.18 μM), followed by Linsitinib ($IC_{50}$ 800 nM), while Ponatinib exhibited little effect

(Supplementary Fig. S1b), indicating that potentially histone-acetylation, event could be involved in abiraterone resistance. Histones were purified from abiraterone-treated C4-2B cells and subjected to mass spectrometry-based identification of post-translational modification. An epigenetic event, acetylation of Lysine at 130th position (K130) in histone H2A was detected (Fig. 1a). The K130 site in histone H2A is evolutionarily conserved in vertebrates, indicating the significance of this modification (Fig. 1b).

To explore the functional importance of K130-acetylated H2A (H2A-K130ac), we generated a high-affinity antibody, which specifically recognized the peptide derived from histone H2A that is K130-acetylated, but not the same peptide without acetylation (Fig. 1c). Further, the ac130-peptide (but not H2A peptide) competed with the antibody for binding, dampening the signal (Supplementary Fig. S1c). In addition, H2A-K130ac antibodies did not cross react with other peptides derived from other histones and their modifications (Supplementary Fig. S1d). Moreover, we generated Myc-tagged constructs expressing either wild-type or K130A and K130S mutants of H2A. HEK293T cells were transfected, and lysates were subjected to immunoprecipitation (IP) with H2A-K130ac antibodies, followed by immunoblotting (IB) with Myc antibodies. Although H2A was K130-acetylated, both the mutants exhibited almost complete loss of acetylation (Fig. 1d). To examine whether H2A-K130ac marks are deposited in temporal manner, abiraterone treated C4-2B, VCaP and LAPC4 cells were subjected to IP with H2A-K130ac antibodies, followed by IB with H2A antibodies. A significant increase in H2A-K130ac levels were seen within 2 to 4 h of abiraterone treatment, which were erased by ~18 h (Fig. 1e and Supplementary Fig. S1e). Prolonged serum starvation causing decrease in androgen levels, also caused a similar effect (Supplementary Fig. S1f), indicating that CRPCs evoke an epigenetic mark, H2A-K130ac in response to androgen deprivation.

### KAT2A acts as epigenetic writer and HDAC1 & 2 as epigenetic erasers of the H2A-K130ac marks

To identify the Histone Deacetylase (HDAC) responsible for regulating H2A-K130ac marks, HDAC1, 2 and 3 expression was knocked down using siRNAs (Supplementary Fig. S2a). A robust increase in H2A-K130ac levels was seen upon HDAC1 and 2 depletion, in contrast, HDAC3 knockdown was not effective (Fig. 1f and Supplementary Fig. S2b), suggesting that HDAC1/2 could be the epigenetic eraser for H2A-K130ac epigenetic marks. To further validate, cells were treated with HDAC1/2 inhibitor, Romidepsin which caused a significant increase in H2A-K130ac levels (Fig. 1g).

To determine the histone acetyl transferase (HAT) responsible for modifying H2A at K130, various HATs were overexpressed, and H2A-K130ac levels were examined, revealing that multiple HATs could target H2A (Supplementary Fig. S1g). Mammalian KAT2A and PCAF exhibit high sequence similarities and have similar biochemical specificities and also possess redundant functions, e.g. both acetylate H3K9, in mouse embryonic fibroblasts (MEFs)[23,24]. To obtain further insight, we used mouse embryo fibroblasts (MEFs) generated from conditional knockout mice; *Kat2a*flox/Δ (Wt expression), *Pcaf*-/-; *Kat2a*flox/Δ (loss of PCAF expression), and, *Pcaf*-/-; *Kat2a*flox/Δ *Cre* (loss of KAT2A & PCAF expression)[23,25]. Immunoblotting revealed a significant decrease in H2A-K130ac marks in MEFs derived from *Pcaf*-/-; *Kat2a*flox/Δ, which showed further loss in *Pcaf*-/-; *Kat2a*flox/Δ *Cre* MEFs (Fig. 1h). Due to highest levels of H2A-K130ac marks accumulation upon KAT2A treatment, KAT2A was used as a model K-130 HAT for subsequent studies.

### Androgen deprivation drives marking of *SREBF1* exons with H2A-K130ac, promoting its transcriptional activation

To decipher the H2A-K130ac epigenetic footprint, chromatin prepared from vehicle- or Romidepsin-treated cells was immunoprecipitated (ChIP) with H2A-K130ac antibody, followed by sequencing (ChIP-seq). Interestingly, the top 20 sites of the H2A-K130ac marks deposition

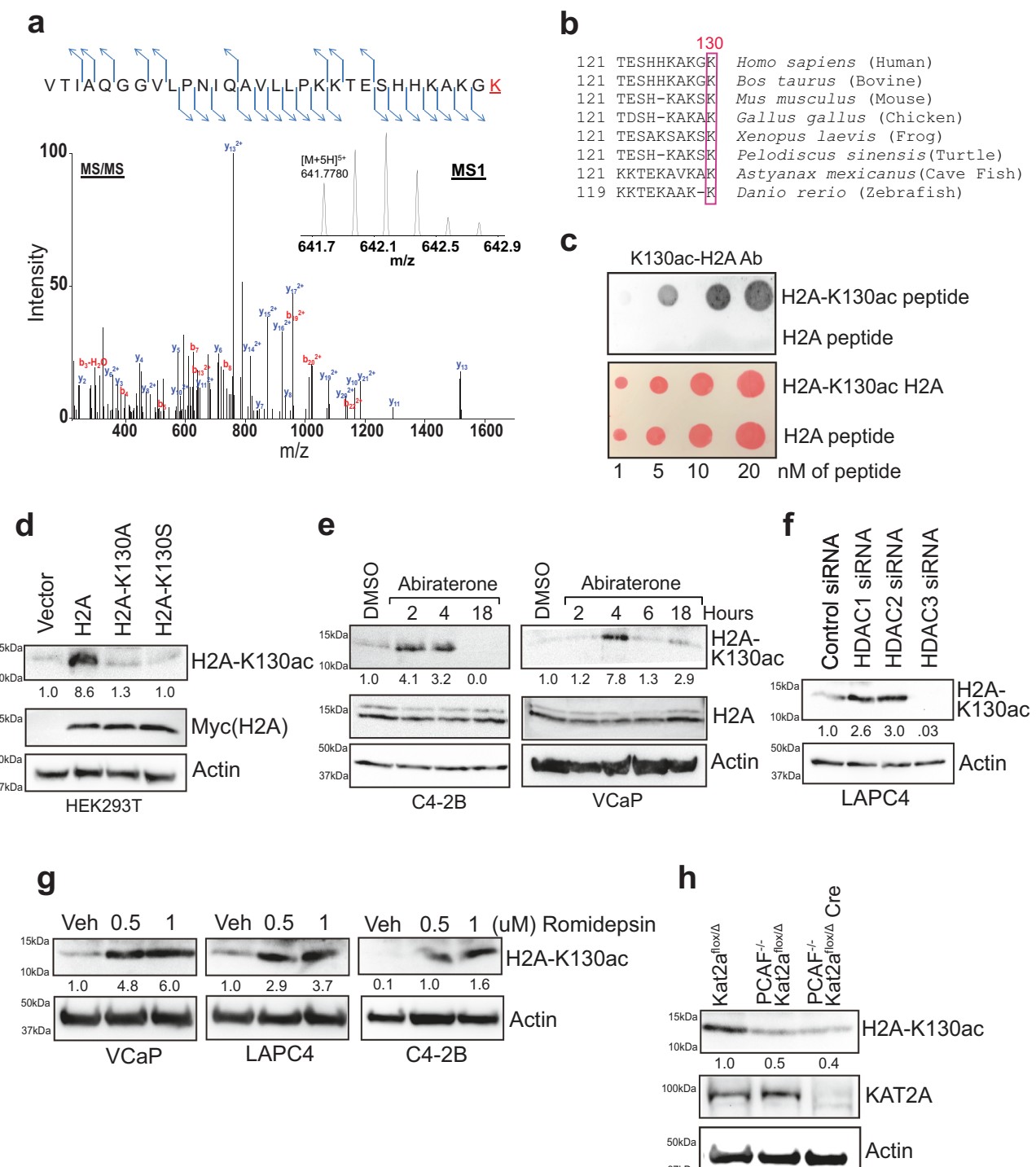

**Fig. 1 | CRPCs enrich epigenetic marks, H2A-K130ac upon androgen deprivation. a** Histones purified from abiraterone-treated C4-2B cells were subjected to mass spectrometry-based identification. The intact peptide (MS1) was detected with *m/z* of 641.7780(5+), which represents a ppm error of 0.70. The MS/MS spectrum was matched to the peptide with Lys130 acetylated VTIAQGGVLPNI-QAVLLPKKTESHHKAKGK. The identification was made by both Sequest and Mascot with Xcorr 6.57, delta CN 0.03, and Mascot ion score of 36. **b** Evolutionary conservation of Lys130 in histone H2A. **c** Acetylated histone H2A peptide and corresponding non-acetylated peptides were immunoblotted (IB) with H2A-K130ac antibody. Lower blot is Ponceau S stained. **d** HEK293 cells were transfected with constructs expressing Myc-tagged H2A or its mutants, K130A and K130S. Lysates were immunoprecipitated (IP) with H2A-K130ac antibody, followed by

immunoblotting (IB) with Myc antibody (top panel). Lysates were also subjected to IB with Myc and Actin antibodies (lower panels). **e** C4-2B and VCaP cells were treated with abiraterone acetate (7.5 μM) in -FBS media for 2, 4, or 18 h. Lysates were IP with H2A-K130ac antibody, followed by IB with H2A antibody (top panel). **f** LAPC4 cells were transfected with siRNAs for HDAC1, 2 and 3 and the lysates were IP with H2A-K130ac antibody, followed by IB with H2A antibody (top panel). **g** VCaP, LAPC4, and C4-2B cells were treated with vehicle or Romidepsin (0.5 & 1 μM), overnight and the lysates were IP with H2A-K130ac antibody, followed by IB with H2A antibody. **h** MEFs were stimulated with insulin for 30 min and lysates were IP with H2A-K130ac antibody, followed by IB with H2A antibody (top panel). Representative images are as shown (*n* = 3 biologically independent experiments). Source data are provided as a Source Data file.

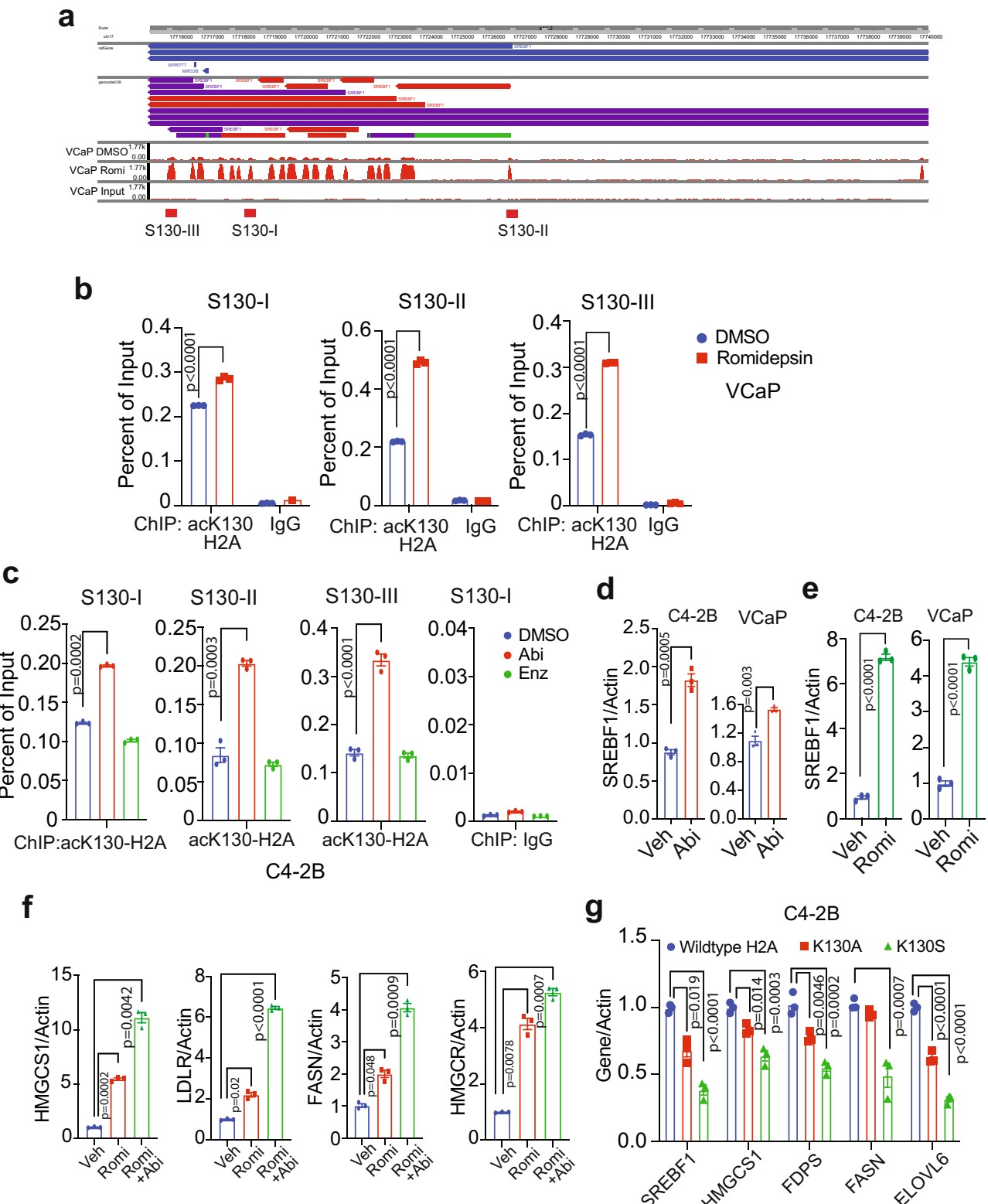

were the 20 exons of *SREBF1* gene (Fig. 2a). The locations of the H2A-K130ac marks deposition are shown in Supplementary Data 1. Venn diagram shows the distribution of peaks (Supplementary Fig. S2c). The H2A-K130ac binding motifs predicted by HOMER, including the corresponding relative score, sequence, and transcription factors are shown in Supplementary Fig. S2d, which show a distinct set of Motifs used by H2A-K130ac, including SREBF1 (bHLH), HNF6B (ONECUT2), GCM1 (Glial Cells Missing Transcription Factor 1) and RNA Pol II.

To verify the H2A-K130ac mark deposition in *SREBF1* gene, ChIP was performed in C4-2B and VCaP cells, followed by real-time PCR (ChIP-qPCR) using 3 site-specific primers, corresponding to exons 2, 14 and 20, named here as **S**REBF1 ac**130** sites **I** to **III** or S130-I to III (Fig. 2a). ChIP with H2A-K130ac antibodies exhibited H2A-K130ac marks deposition at all the three sites, S130I-III, however, ChIP with IgG did not exhibit binding at these sites (Fig. 2b and Supplementary Fig S2e). Further, upon Romidepsin treatment, there was a significant

**Fig. 2 | Deposition of H2A-K130ac epigenetic marks in the exons of *SREBF1* gene augments its expression. a** Romidepsin (0.5 μM, 18 h) treated VCaP cells were cross-linked, chromatin was isolated, and ChIP was performed using H2A-K130ac antibody or IgG, followed by sequencing. The peaks in the exons of *SREBF1* gene and primers' locations, designated as S130-I/II/III are shown in graphical format. **b** ChIP was performed as described above using H2A-K130ac antibody or IgG, followed by qPCR using primers corresponding to S130-I/II/III. **c** Abiraterone (Abi) or Enzalutamide or Enz (7.5 μM, 18 h) treated C4-2B cells were cross-linked and ChIP was performed using H2A-K130ac antibody, followed by qPCR using primers corresponding to S130-I/II/III. **d** RNA isolated from C4-2B and VCaP cells treated with vehicle or abiraterone (7.5 μM, 18 h) and was subjected to qRT-PCR with *SREBF1* and actin primers. **e** RNA isolated from C4-2B and VCaP cells treated with vehicle or Romidepsin (0.5 μM, 18 h) and was subjected to qRT-PCR with *SREBF1* and actin primers. **f** RNA isolated from C4-2B cells treated with vehicle, Romidepsin (0.5 μM, 18 h) or Romidepsin & abiraterone (0.5 μM and 7.5 μM) and qRT-PCR was done with indicated primers. **g** RNA isolated from C4-2B cells transfected with constructs expressing wildtype H2A or mutants K130A-H2A, K130S-H2A and was subjected to qRT-PCR with indicated primers. For (**b**–**g**) The experiments were performed in triplicates (*n* = 3), and the experiments were repeated thrice independently with similar results; a representative dataset is shown. Data are represented as mean ± SEM. For (**b**–**e**), *p* values were determined by unpaired two-tailed Student's *t* test. For (**f**, **g**), *p* values were determined by one-way ANOVA. Source data are provided as a Source Data file.

increase in H2A-K130ac deposition in *SREBF1* gene (Fig. 2b and Supplementary Fig. S2e).

To validate deposition of the H2A-K130ac marks in *SREBF1* gene is caused due to loss of androgen, C4-2B cells were treated with abiraterone and ChIP-qPCR was performed. In addition, cells were also treated with another AR-antagonist, enzalutamide, which unlike Abiraterone does not affect androgen synthesis, but instead inhibits AR nuclear translocation[26]. Abiraterone significantly increased H2A-K130ac marks deposition in *SREBF1* gene, however, enzalutamide treatment caused no change in H2A-K130ac marks deposition (Fig. 2c), suggesting that the reduction in androgen levels promotes deposition of H2A-K130ac epigenetic marks. To assess the outcome of H2A-K130ac epigenetic marks deposition, *SREBF1* mRNA levels were examined. Treatment with abiraterone and Romidepsin resulted in robust increase in *SREBF1* mRNA expression (Fig. 2d, e and Supplementary Fig. S2f). To determine consequent SREBF1 targeted cholesterologenic and lipogenic gene signature, C4-2B cells were treated with Romidepsin or Romidepsin & abiraterone and subjected to qRT-PCR. A significant increase in *SREBF1* target genes, *HMGCS1, LDLR, FASN* and *HMGCR* was seen upon Romidepsin treatment, which was further increased upon Romidepsin & Abiraterone treatment (Fig. 2f and Supplementary Fig S2g).

To further validate the role of H2A-K130ac epigenetic marks, C4-2B cells were transfected with the Flag-tagged constructs expressing either wild-type or K130A and K130S mutants of H2A. *SREBF1* and target cholesterologenic and lipogenic genes (*HMGCS1, FDPS, FASN,* and *ELOVLS*) expression was significantly compromised in mutant expressing cells (Fig. 2g and Supplementary Fig. S2h–k). Together, these data indicate that androgen deprivation induces H2A-K130ac epigenetic marks deposition in *SREBF1*, causing its transcriptional activation, initiating cholesterologenic and lipogenic transcription program.

## SREBF1 Y673 & Y951-phosphorylation and its nuclear translocation in androgen deficient environment

We reasoned that deposition of H2A-K130ac epigenetic marks can best be regulated by a protein that 'senses' androgen levels. Earlier, we observed that the abiraterone-resistant cells were sensitive to IR inhibitor, Linsitinib (Supplementary Fig. S1b), opening the possibility that a hitherto unknown Tyr-phosphorylation event is involved in abiraterone-resistance. To identify the event, abiraterone-treated C4-2B cells were immunoprecipitated using pTyr-beads, followed by mass spectrometry-based identification of the bound protein. Surprisingly, a Tyr-phosphorylated form of SREBF1 was identified with two Tyr-phosphorylation sites, Y673 and Y951 (Y649 & Y927 in SREBP-1C, and Y703 & Y981 in longer form) (Supplementary Fig. S3a, b, also see Fig. 3a). Y673 site is conserved in vertebrates; however, Y951 sites is conserved in mammals, but not in amphibians and fishes (Supplementary Fig. S3c).

To examine the relevance of SREBF1 activation in response to abiraterone, VCaP, LAPC4, and C4-2B cells were treated with abiraterone and fractionation was performed, followed by immunoblotting.

A 68kDA nSREBF1 (N-terminal cleaved form SREBF1 with bHLH-LZ, see Fig. 3a) was specifically enriched in the nuclear fraction upon abiraterone treatment (Fig. 3b). Further to examine role of SREBF1-phosphorylation, Y673/951A double mutant was generated, which was tagged at both the ends; HA-tag at N-terminus and His (& Myc)-tag at C-terminus (Fig. 3a). SREBF1 double mutant, Y673/951A, exhibited almost complete loss of nuclear localization (Fig. 3c). In addition, robust endogenous SREBF1 Tyr-phosphorylation was observed upon androgen deprivation in C4-2B, LAPC4 and VCaP cells (Fig. 3d, e and Supplementary Fig. S4a). Together these data indicate that SREBF1 phosphorylation at Y673 and Y951 sites is needed for its nuclear localization in androgen-deficient environment.

To identify the kinase/s responsible, HEK293T cells were co-transfected with constructs expressing various kinases and SREBF1, revealing that FGFR and HER4 kinases are capable of SREBF1 Tyr-phosphorylation (Supplementary Fig. S4b). To confirm Tyr-phosphorylation of endogenous SREBF1, C4-2B cells were treated with FGF and insulin, which too exhibited robust SREBF1 Tyr-phosphorylation in temporal manner (Supplementary Fig. S4c and d). Further, cells were transfected with constructs expressing SREBF1 or double mutant, Y673/951A with these kinases, revealing almost complete loss of phosphorylation in the double mutant (Fig. 3f–h), validating that SREBF1 is primarily phosphorylated at Y673 & Y951 sites by FGFR, HER4 and IR kinases.

## nSREBF1 recruited KAT2A and marked *SREBF1* locus with H2A-K130ac, causing transcriptional activation of SREBF1 and its target genes

To determine the role of SREBF1-phosphorylation, androgen-deprived cells were fractionated, followed by immunoblotting. Abiraterone treatment exhibited detection of nSREBF1 in both *cis* and *trans* Golgi; however, Y673/951A double mutant was distinctly lacking (Fig. 4a), suggesting that phosphorylation is needed for SREBF1 translocation into Golgi in response to androgen-deprivation. Further, a significant increase in H2A-K130ac levels was seen in SREBF1 expressing cells upon androgen deprivation; however, Y673/951A double mutant expressing cells exhibited marked decrease in H2A-K130ac levels (Fig. 4b, c). To assess whether phosphorylation of SREBF1 is needed for its chromatin binding, C4-2B cells were transfected with HA-tagged SREBF1 or Y673/951A double mutants (empty vector as control), treated with abiraterone, and ChIP was performed with HA antibodies, followed by qPCR with primers corresponding to *SREBF1* exon 2 (S130-II). A robust nSREBF1 binding was seen in the presence of abiraterone; however, phosphorylation-deficient double mutant Y673/951A exhibited a significant decrease in binding (Fig. 4d). These data suggest that in androgen-deficient environment, SREBF1 Y673/951-phosphorylation is not only needed for its ER to Golgi transport, but also plays a crucial role in its subsequent nuclear localization, and deposition of H2A-K130ac epigenetic marks.

To further validate crucial role of SREBF1 Y673&951-phosphorylation, RNA was isolated from C4-2B and VCaP cells infected with retroviruses expressing SREBF1 and the Y673/95AF double mutant.

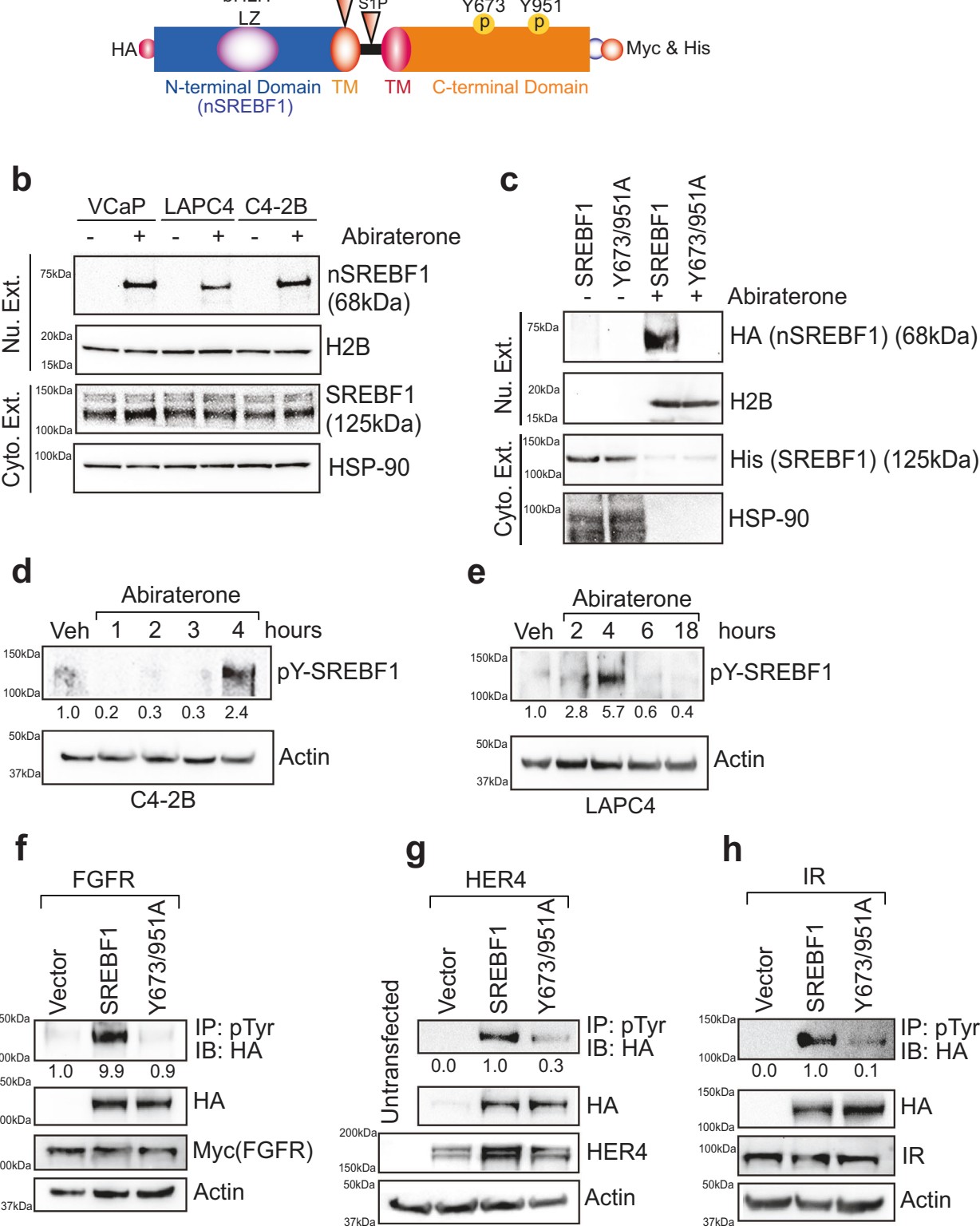

A significant decrease in *SREBF1* and its target gene expression was seen in Y673/951A double mutant expressing cells, in abiraterone-rich condition (Fig. 4e and Supplementary Fig. S4e). To examine the consequence of SREBF1-phosphorylation, growth factors (serum) and androgen-deprived C4-2B cells were treated with insulin or FGF ligands and mRNA levels were determined. A significant increase in mRNA expression of *SREBF1* and its target cholesterologenic

and lipogenic genes was seen upon ligand treatments (Supplementary Fig. S4f).

Based on our earlier observation that KAT2A acts as a HAT for H2A-K130ac (Fig. 1h), we postulated that nSREBF1 could potentially recruit KAT2A to deposit these marks. Co-IP revealed abiraterone dependent SREBF1-KAT2A complex formation (Fig. 4f). Further, SREBF1-KAT2A complex formation in the presence of abiraterone was significantly

**Fig. 3 | Phosphorylation of SREBF1 at Y673 and Y951 promotes its nuclear translocation in androgen deficient environment. a** Representation of SREBF1 expressing construct with N-terminal HA-tag, C-terminal Myc/His tags, and Tyr-phosphoryaltion sites. S1P & S2P are proteolytic cleavage sites and TM represents transmembrane domains. **b** VCaP, LAPC4, and C4-2B cells were treated with abiraterone (3 μM) for 4 h and fractionation was performed, followed by immunoblotting of nuclear and cytosolic fractions with indicated antibodies. **c** HEK293T cells were transfected with SREBF1 or Y673/951A double mutant, followed by abiraterone treatment (3 μM) for 4 h. Fractionation was performed,

followed by immunoblotting of nuclear and cytosolic fractions with indicated antibodies. **d**, **e** C4-2B and LAPC4 cells were treated with abiraterone (3 μM) in -FBS media for indicated time. Lysates were immunoprecipitated with SREBF1, followed by immunoblotting with pTyr antibody. **f–h** HEK293T cells were co-transfected with FGFR, HER4 or IR and HA-tagged SREBF1 or Y673/951A double-mutant. Lysates were immunoprecipitated with pTyr antibody on beads, followed by immunoblotting with anti-HA antibody (top panel). Representative images are as shown (*n* = 3 biologically independent experiments). Source data are provided as a Source Data file.

compromised in phosphorylation deficient Y673/951A double mutant (Fig. 4g and Supplementary Fig. S5a). We purified FLAG-tagged KAT2A and MYC-tagged N-SREBF1 from HEK293T transfected cells and assessed their interaction ex vivo (Supplementary Figure 5b). It appears that KAT2A could interact with N-SREBF1 in DNA independent manner.

We also noticed PCAF binding to SREBF1 (Supplementary Fig. S5c). However, increased H2A-K130ac levels upon Abiraterone treatment in the presence of KAT2A, was not seen in PCAF expressing cells (Supplementary Fig. S5d, compare lanes 1 to 2 and 3 to 4). This does not completely rule out the potential role of PCAF in deposition of H2A-K130ac marks in abiraterone dependent manner, as multiple HATs could be involved H2A-K130ac marks deposition depending on cell type or disease stage. Moreover, KAT2A binding to *SREBF1* locus was significantly reduced when phosphorylation deficient Y673/951A double mutant was expressed (Fig. 4h and Supplementary Fig. S5e). Thus, androgen deficiency induces an auto-activating loop wherein Y673/951-phosphorylated SREBF1 undergoes efficient nuclear translocation; the nuclear SREBF1 recruits KAT2A to *SREBF1* locus to deposit H2A-K130ac marks to activate SREBF1 expression and initiate cholesterologenic and lipogenic transcription program.

### SREBF1 Y673/951-phosphorylation is required for Abiraterone-resistant CRPC Tumor Growth

To investigate role of pY673/951-SREBF1 in CRPC growth in abiraterone-rich environment, C4-2B cells were retrovirally infected with SREBF1 or Y673/951A double mutant followed by treatment with increasing concentration of abiraterone. As expected SREBF1 expressing C4-2B cells proliferated even at high concentration of abiraterone (IC$_{50}$ 6.3 μM), in contrast, Y673/951A double mutant expressing cells exhibited considerable sensitivity to abiraterone (IC$_{50}$ 0.73 μM), indicating SREBF1 Y673 & Y951-phosphorylation is critical for abiraterone-resistance (Fig. 5a). To validate these findings, $1.2 \times 10^6$ C4-2B cells retrovirally infected with SREBF1 or Y673/951A double mutant were injected in castrated SCID mice. Once palpable tumors were noticed (5 weeks), mice were orally gavaged with abiraterone and formation of tumor was monitored. SREBF1 expressing C4-2B formed robust tumor growth reaching ~1300 mm$^3$ in 9 weeks, in contrast, Y673/951A double mutant expressing cells exhibited severely compromised tumors growth (Fig. 5b). At the end of the experiment, mice were humanely euthanized, and tumors were excised, which revealed formation of tumors in 2 out of 7 mice that were injected with Y673/951A expressing cells. However, all 6 mice that were injected with SREBF1 expressing cells formed robust tumor growth (Fig. 5c, d). Weights of the mice were also monitored during this experiment, which showed decrease in weights of mice that were injected with SREBF1 expressing cells (and formed robust tumors), in contrast, mice that were injected with Y673/95AF expressing cells barely had any tumors and did not exhibit any weight loss (Fig. 5e).

To determine whether the decrease in CRPC tumor growth is due to significant loss of H2A-K130ac and SREBF1 expression, immunohistochemical (IHC) staining xenograft tumors was performed. A significant loss of H2A-K130ac epigenetic marks and SREBF1 expression was observed in Y673/951A double mutant expressing tumors, in contrast, high levels of both of these were seen in SREBF1 expressing tumors (Fig. 5f). Based on these data we postulated that synchronous

inhibition of KAT2A and Tyr-kinases targeting SREBF1 phosphorylation could overcome abiraterone-resistance. To address this possibility in vivo, TRAMP-C2 cells were used which form CRPC tumors[27]. TRAMP-C2 cells were injected in castrated C57BL/6 (B6) mice and once palpable tumors were formed, mice were gavaged with abiraterone or combinatorial treatment consisting of GCN5 inhibitor, CPTH2, and HER4 inhibitor, Afatinib. Tumor growth continued upon abiraterone treatment; however, significant loss of tumor growth was observed upon administration of CPTH2+Afatinib inhibitor combination (Fig. 5g).

To further demonstrate the efficacy of CPTH2, and Afatinib inhibitor combination in human CRPCs, abiraterone-resistant C4-2B cells were injected in SCID mice and once palpable tumors were formed, mice were gavaged with abiraterone or CPTH2+Afatinib inhibitor combination. C4-2B tumor growth was unaffected by abiraterone, however, administration of CPTH2+Afatinib combination significantly compromised tumor growth (Fig. 5h, i). Taken together, these data indicate that pY673/951-SREBF1/H2A-K130ac signaling is crucial for abiraterone-resistance and abiraterone-resistant CRPCs can be sensitized by CPTH2+Afatinib inhibitor combination.

### pY-SREBF1 senses androgen deficiency by dissociating from AR

We reasoned that SREBF1 phosphorylation and nuclear translocation are dependent on androgen levels, which it sensed through its interaction with androgen bound AR. To test this hypothesis, HEK293T cells were transformed with FLAG-tagged AR, treated with Dihydrotestosterone (DHT), and co-immunoprecipitation studies were performed. Robust SREBF1-AR complex formation was noticed upon DHT treatment (Fig. 6a). A similar DHT dependent endogenous SREBF1-AR complex formation was observed in VCaP and C4-2B cells (Fig. 6b and Supplementary Fig. S6a). Further, co-immunoprecipitation studies revealed an increase in AR binding with Y673/951A double mutant in androgen-rich environment (Fig. 6c), suggesting that phosphorylation of SREBF1 at Y673/951 is necessary for its dissociation from AR. Together, these data indicate that androgen-bound AR forms a complex with SREBF1 retaining it in cytoplasm; however, when deprived of androgen, SREBF1 undergoes Y673/951-phosphorylation and dissociates from androgen-unbound AR, promoting its nuclear translocation.

### Tumor-derived androgen promote T cell exhaustion by subduing H2A-K130ac, promoting abiraterone-resistant CRPC growth

Prostate cancers are highly dependent on androgen receptor (AR) signaling[19,20,22], and are refractory to immune checkpoint blockade (ICB)[28–30]. Recent studies have shown that inhibition of AR activity by enzalutamide in CD8$^+$ T cells prevented T cell exhaustion and improved responsiveness to PD-1 targeted therapy[31]. Interestingly, combining enzalutamide with the PD-1 inhibitor, pembrolizumab had a modest response in CRPCs; only 5 out of 28 patients (18%), had a PSA decline of ≥50%[32]. We postulated that continuous non-inflamed nature of CRPCs is due to paracrine effect of tumor-derived androgen on T cells. CD8$^+$ T cells have robust AR expression opening a possibility that androgen may contribute to dampening of CD8$^+$ T cells[33,34]. To explore whether tumor-derived androgen regulates pSREBF1/H2A-K130ac signaling in T cells, TRAMP-C2 cells were implanted in C57BL/6

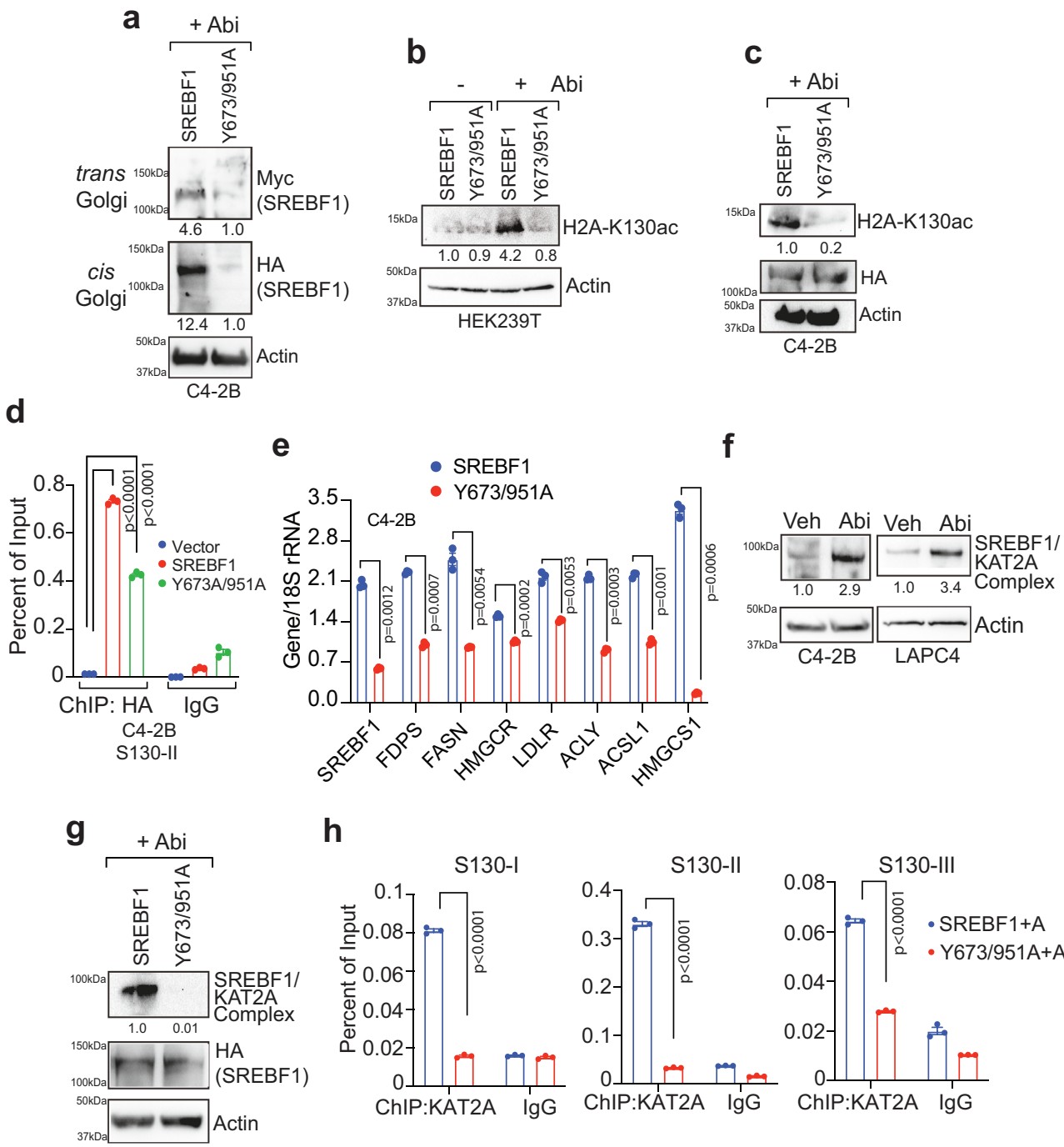

**Fig. 4 | nSREBF1 recruits KAT2A to mark *SREBF1* locus and promotes its transcription. a** C4-2B cells were infected with SREBF1 and Y673/951A double mutant expressing constructs, followed by abiraterone treatment (7.5 μM) for 4 h. The lysates were fractionated into trans and cis Golgi fractions and were immunoprecipitated with SREBF1 antibody, followed by immunoblotting with Myc or HA antibody. **b** HEK293T cells were transfected with SREBF1 and Y673/951A double mutant, followed by abiraterone treatment (7.5 μM) for 4 h. The lysates were immunoprecipitated with H2A-K130ac antibody, followed by immunoblotting with H2A antibody. **c** C4-2B cells were transfected with HA-tagged SREBF1 or Y673/951A double-mutant, followed by abiraterone treatment (7.5 μM) for 4 h. Lysates were immunoprecipitated with H2A-K130ac antibody, followed by immunoblotting with H2A antibody. **d** C4-2B cells were infected with SREBF1 or Y673/951A double mutant expressing constructs. Cells were treated with abiraterone (7.5 μM) for 4 h, followed by ChIP with HA (or IgG) antibodies. PCR was performed using S130-II primers.

**e** RNA isolated from C4-2B cells infected with SREBF1 or Y673/951A double mutant expressing constructs and subjected to qRT-PCR with indicated primers. **f** C4-2B and LAPC4 cells were treated with abiraterone (7.5 μM) and lysates were immunoprecipitated with SREBF1, followed by immunoblotting with KAT2A antibody. **g** C4-2B cells were infected with HA-tagged SREBF1 or Y673/951A double-mutant expressing constructs. Lysates were immunoprecipitated with anti-HA affinity gel followed by immunoblotting with KAT2A antibody. **h** C4-2B cells were infected with SREBF1 or Y673/951A double mutant expressing constructs. Cells were treated with abiraterone (7.5 μM) for 4 h, followed by ChIP with KAT2A (or IgG) antibodies. PCR was performed using S130-1/II/III primers. For (**d**, **e**, **h**) (*n* = 3), the experiments were repeated thrice independently with similar results, and a representative dataset is shown. Data are represented as mean ± SEM. For (**d**), *p* values were determined by one-way ANOVA. For (**e**, **h**), *p* values were determined by paired and unpaired two-tailed Student's *t* test, respectively. Source data are provided as a Source Data file.

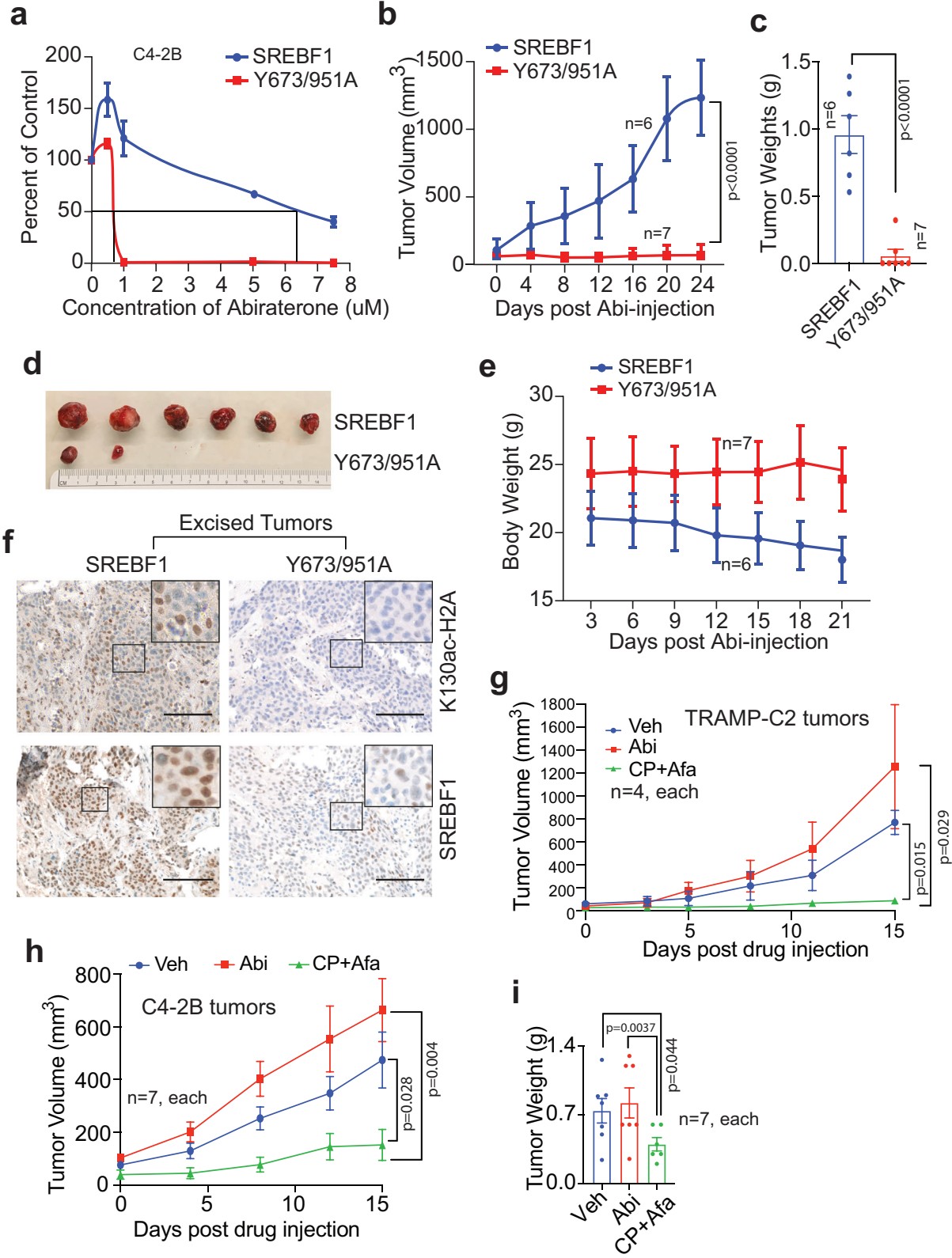

mice. TRAMP-C2 cells develop CRPC tumors in the absence of testes-derived androgens (Fig. 5g and h). 5 weeks post-injection of TRAMP-C2 cells, when tumors were palpable, the mice were given oral gavage with abiraterone or CPTH2+Afatinib for 4 weeks. To determine whether TRAMP-C2 cells resorted to de novo androgen synthesis in the presence of abiratetone, androgen levels in blood serum were measured. abiraterone treated mice exhibited a significant increase in serum

androgen levels (Fig. 6d), indicating that abiraterone treatment kick-start de novo androgen synthesis in prostate tumors. Presence of tumor-derived androgen resulted in robust tumor growth in mice treated with abiraterone; however, CPTH2+Afatinib inhibitor treated mice exhibited striking tumor reduction (Fig. 6e and Supplementary Fig. S6b). Further, abiraterone treated tumors exhibited significant increase in overall pY-SREBF1 and H2A-K130ac expression, which was

**Fig. 5 | Phosphorylation-deficient SREBF1 expressing tumors are sensitive to abiraterone treatment and CPTH2+Afatinib combination therapy overcomes abiraterone resistance. a** C4-2B cells were infected with the retroviral constructs encoding SREBF1 or Y673/951A double-mutant constructs. Cells were treated with indicated concentration of abiraterone for 96 h. The number of viable cells were counted by trypan blue exclusion assay. The experiment was performed in triplicates and two biological replicates. **b** C4-2B cells described above were implanted subcutaneously in male SCID mice with n = 6 in WT SREBF1 group and n = 7 mice in the double mutant group. When tumors became palpable, mice were orally gavaged with abiraterone for twice a week; total 8 doses were performed. Tumor volumes were measured with calipers. **c** Tumors were excised, and weights of the tumors are shown. **d** A photograph of the tumors is shown. **e** Body weights of the SCID mice during the treatment period are shown. **f** Immunohistochemical staining

of the excised tumors was performed using SREBF1 and H2A-K130ac antibodies. Representative images from 6 samples of WT SREBF1 group and 7 samples of double mutant group are shown. Scale bar 200 μM. **g** TRAMP-C2 cells were implanted subcutaneously in C57BL/6 mice. When tumors became palpable, mice were orally gavaged with abiraterone or CPTH2+Afatinib for five days a week, for 3 weeks (n = 4 mice in each arm). Tumor volumes were measured with calipers. **h** C4-2B tumor bearing male SCID mice were orally gavaged with vehicle (10% DMSO) or abiraterone or CPTH2+Afatinib for five days a week, for 3 weeks (n = 7 mice in each group). Tumor volumes were measured with calipers. **i** Tumors were excised, and weights of the tumors are shown. Data are represented as mean ± SEM. For (**b, c, g–i**), p values were determined by unpaired two-tailed Student's t test, respectively. Source data are provided as a Source Data file.

compromised upon CPTH2+Afatinib treatment (Fig. 6f). It was also reflected significant loss of *Srebf1, Fasn, Ldlr1*, and *Hmgcs1* expression in tumors derived from CPTH2+Afatinib treated mice (Fig. 6g), suggesting that inhibition of pY-SREBF1/H2A-K130ac signaling suppresses de novo androgen synthesis in tumors.

To assess paracrine effect of tumor-derived androgen, spleens were harvested, which revealed a significant decrease in CD3[+] T cells when blood androgen levels were high (upon abiraterone treatment), in contrast, CPTH2+Afatinib treated mice exhibited increased CD3[+] T cells (Fig. 6h and Supplementary Fig. S6c). Consistent with these data, the exhaustion markers, PD-1 and Lag3, were significantly decreased in effector CD8[+] T cells obtained from tumor draining lymph nodes of CPTH2+Afatinib treated mice (Fig. 6i, j and Supplementary Fig. S6d). In addition, a significant increase in the interferon-γ levels was observed in the blood serum of CPTH2+Afatinib treated mice (Fig. 6k). Further, to examine the consequences of the androgen treatment on the ability of T cells to induce calcium mobilization, we examined the calcium flux in splenocytes isolated from C57BL/6 mice and treated with vehicle or DHT for 1 h. Splenocytes treated with anti-CD3 antibody exhibited a robust calcium response, however it was significantly dampened in those cells that were DHT treated (Fig. 6l). To determine the role of androgen in T cell exhaustion, Jurkat cells were treated with DHT and nuclear/cytoplasmic fractionation was performed. Post 60 min of DHT treatment, there was barely any nuclear SREBF1; however, cytosolic SREBF1 levels were significantly increased (Fig. 6m). Taken together these data demonstrate that androgen effectively compromise nuclear translocation of SREBF1, leading to marked immunosuppression by promoting exhaustion of T cells.

## H2A-K130ac and nSREBF1 expression correlate with disease progression in metastatic prostate cancer

To determine the clinical relevance of H2A-K130ac mark deposition on transcriptionally active SREBF1 (nuclear) expression, we performed immunohistochemical staining of clinically annotated prostate tissue microarrays (n = 80 samples) with respective antibodies. The H2A-K130ac and SREBF1 (nuclear) levels increased as cancer progressed from the stage I to IV (Fig. 7a–c). In addition, H2A-K130ac and SREBF1 expression were found to positively correlate with each other during disease progression (Pearson correlation r = 0.66, p < 0.0001) (Fig. 7d). Also see Supplementary Data 2. Collectively, these data indicate that the expression of H2A-K130ac, and SREBF1 progressively increase as disease progressed to later stages, with the highest expression in stage IV. Further, it indicates that H2A-K130ac marks may have a direct impact on nuclear SREBF1 expression in stage IV prostate cancer.

## Global lipidomic profiling reveal renewed triglycerides and phosphatidylinositol synthesis upon androgen deprivation similar to AA prostate cancer patients

As compared to European American ancestry (EA) men, AA men are twice likely to be diagnosed with prostate cancer, and more than twice

as likely to die of their disease[35]. Increased cholesterol was shown to be one of the risk factors for prostate cancer recurrence in black, but not non-black men[36,37]. Distinct metabolic alterations is emerging to be the causative factor for the racial disparity[38], however identity of metabolites and impetus for their differential synthesis is not clear. We performed unbiased global lipidomic profiling of C4-2B and VCaP cells treated (or untreated) with abiraterone using triple time of flight (Triple-TOF) liquid chromatography-mass spectrometry (LC-MS). Total 16 experimental and 3 liver control samples were used, which led to identification of 1314 metabolites from 2 methods; Pos_Method was normalized by Internal Standard 18:1(d7) Lyso PC (ISTD) and Neg_Method, which was normalized by Internal Standard 15:0-18:1(d7) PE (ISTD) (Supplementary Fig. S7a–g). Comparative levels of metabolites were assessed by T-test and ANOVA (Supplementary Data 2, 3, and 4). Principal component analysis using all the lipids revealed a distinct separation of the lipid components upon androgen deprivation (Supplementary Fig. S8, Supplementary Data 5 and 6). Higher levels of phosphatidyl glycerides (PGs), phosphatidyl cholines (PC), lyso-phosphatidyl cholines (lyso-PC), phosphatidyl inositols (PI), plasmenyl PCs and phosphatidyl ethanolamines (PE) were observed in both VCaP and C4-2B cells treated with abiraterone (Supplementary Fig. S8). In addition, levels of polyunsaturated long chain triglycerides (TGs) and diglycerides (DGs) were specifically elevated in C4-2B and VCaP cells treated with abiraterone, respectively, as compared to their respective untreated control counterparts (Supplementary Fig. S8–10).

We compared these data for altered lipids within CRPC that could distinguish prostate cancer in EA men vs. AA men. Earlier, we had observed a significant elevation in the level of individual lipids belonging to the CE, TG, PI, and PG class of lipids (predominantly polyunsaturated and long chain) in AA men as compared to EA[39], which are also significantly elevated in CRPCs upon androgen deprivation (Supplementary Fig. S9, S10). In addition, the elevated levels of polyunsaturated long chain TGs were seen in C4-2B treated with abiraterone, which is consistent with elevated levels of TG in plasma, and its association with prostate cancer risk and poor clinical outcome in AA men[36,37]. Overall, the similarity in the lipid profile of androgen-deprived CRPCs and AA prostate cancer patients indicates that the pY-SREBF1/H2A-K130ac signaling could play a role in race-specific deregulation of lipid pathways in aggressive prostate tumors.

## Discussion

In contrast to prostate cells, the prostate cancers exhibit a distinct shift towards de novo lipid biosynthesis and steroidogenesis, thus minimizing dependence on the testicular androgen for their survival. This is one of the major reasons behind almost universal resistance seen in abiraterone treated cancer patients. How the prostate cancer cells sense androgen deficiency to switch on a distinct steroidogenesis program was not clearly understood. Serendipitously, we encountered two distinct modification events, (i) histone H2A acetylation at the terminal Lys130 residue, and (ii) Tyr-phosphorylation of SREBF1 at Y673 and

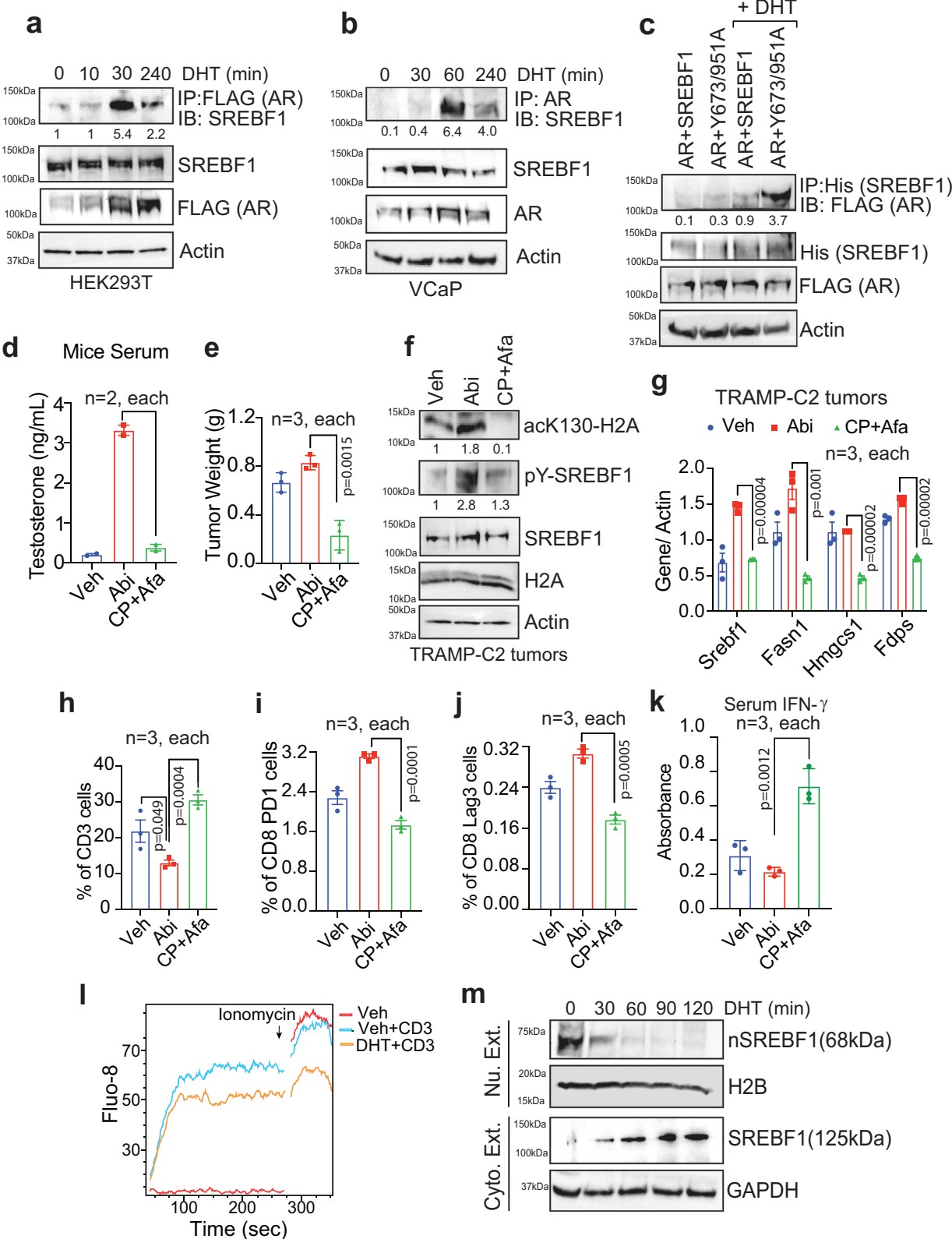

Y951. Although discrete, these two events converged into a single outcome; dual-phosphorylated SREBF1 'sensed' the lack of androgen, and triggered deposition of epigenetic event, H2A-K130ac in *SREBF1* genetic locus to initiate cholesterologenic program in tumor cells (Fig. 8). Interestingly, dual-phosphorylated SREBF1 not only sensed androgen deficiency, but also ensured uninterrupted cholesterol supply for androgen synthesis by able to translocate from ER to Golgi, even in presence of high cholesterol levels. Thus, it created a unique multi-step feed-forward loop thereby SREBF1 accomplishes all the three aspects- sensing of androgen deficiency, recruitment of epigenetic marker causing its own transcriptional upregulation, and boosting androgen production by regulating cholesterologenic program.

Considering the clinical importance of high rates of lipogenesis in rapidly proliferating cancer cell, SREBF1 was postulated to be

**Fig. 6 | SREBF1/AR interaction and its role in androgen sensing. a** HEK293T cells were transfected with FLAG-tagged AR, followed by DHT treatment for indicated time. The lysates were immunoprecipitated with FLAG antibody, followed by immunoblotting with SREBF1 antibody. **b** VCaP cells were DHT treated for indicated time. The lysates were immunoprecipitated with AR antibody, followed by immunoblotting with SREBF1 antibody. **c** HEK293T cells were transfected with FLAG-tagged AR and His-tagged SREBF1 and Y673/951A double mutant constructs, followed by DHT treatment. The lysates were immunoprecipitated with Ni-NTA beads (His), followed by immunoblotting with FLAG antibody. **d** TRAMP tumor bearing C57BL/6 mice orally gavaged with abiraterone or CPTH2+Afatinib for five days a week, for 4 weeks. The blood was collected, and serum androgen levels were determined. **e** Tumors were excised from mice and weighed (n = 3 tumors in each arm). **f** Tumors were excised, and the lysates were immunoprecipitated with H2A-K130ac antibody, followed by immunoblotting with H2A antibody (top panel). Also, lysates were immunoprecipitated with SREBF1, followed by immunoblotting with

pTyr antibody (second panel). **g** RNA isolated from TRAMP-C2 tumors and were subjected to qRT-PCR with indicated primers. (n = 3, 2 replicates). **h** Flow cytometric analysis of splenocytes isolated from tumor bearing mice was done to assess CD3[+] T cell populations (n = 3 mice in each group). **i, j, h** Flow cytometric analysis of lymphocytes from tumor draining lymph nodes of mice implanted with TRAMP-C2 cells was done to assess CD8 populations with exhaustion markers (n = 3 mice in each group). **k** Serum interferon-γ levels were assessed by ELISA (n = 3 mice in each group). **l** Representative calcium flux of splenocytes isolated from C57BL/6 mice treated with vehicle or DHT for 1 h. **m** Jurkat cells were treated with DHT for indicated time, followed by fractionation. Immunoblotting of nuclear and cytosolic fractions was performed with indicated antibodies. For (**a**–**c, f**), and (**m**) representative images are as shown (n = 3 biologically independent experiments). For (**e, g**–**k**), data are represented as mean ± SEM. p values were determined by unpaired two-tailed Student's t test, respectively. Source data are provided as a Source Data file.

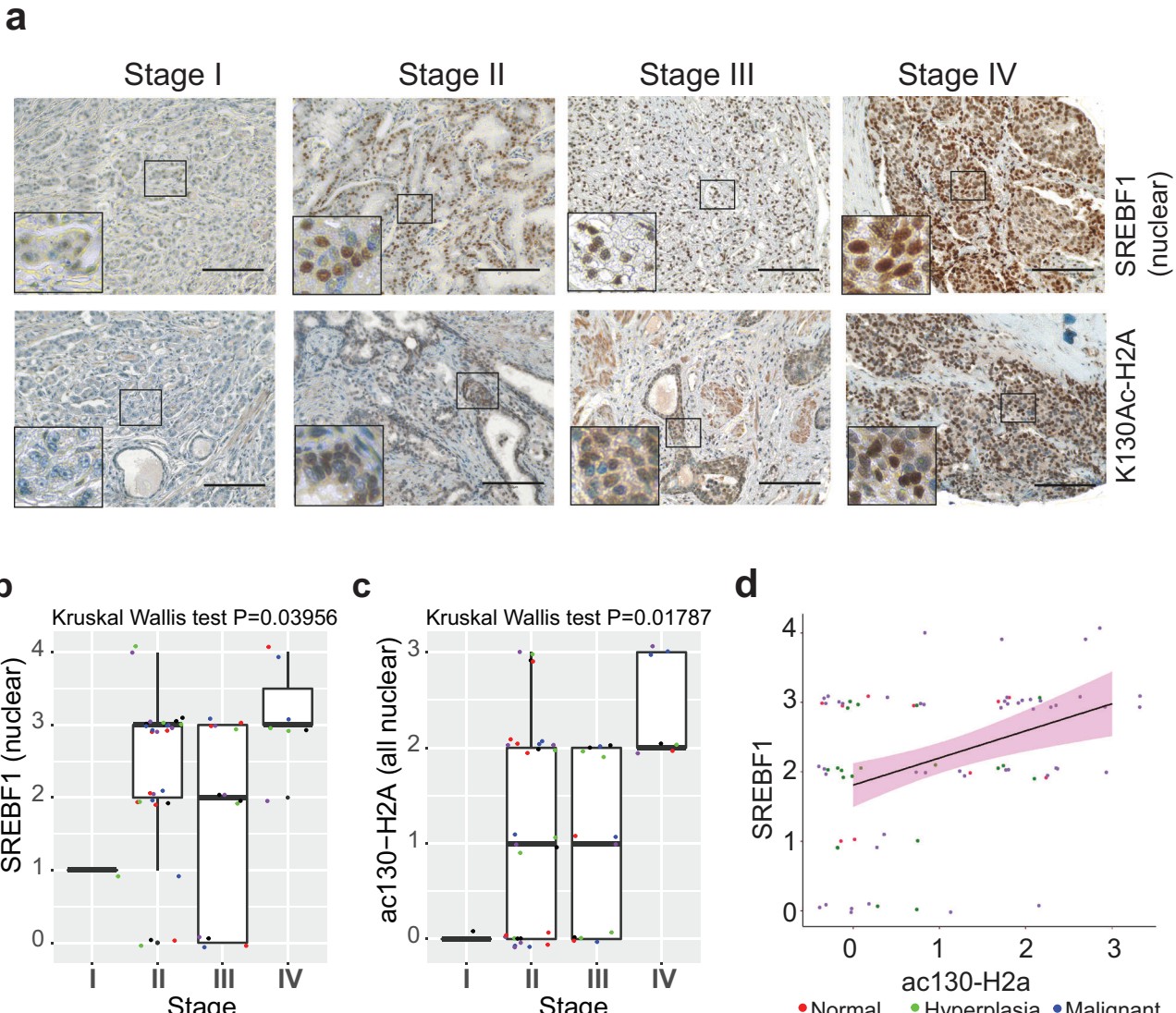

**Fig. 7 | Increased levels of nuclear SREBF1 and H2A-K130ac epigenetic marks in stage IV prostate cancer. a** Representative images of prostate tissue microarray (TMA) sections immunohistochemical stained with SREBF1 and H2A-K130ac antibody (n = 80 cores per slide). Scale bar 200 μM. **b, c** Box plots summarizing distributions of staining intensities for H2A-K130ac and SREBF1 (nuclear) in different stages of prostate cancer, Stage I to IV. Each box has lines at the lower quartile (25%), median (50%), and upper quartile values (75%). Whiskers extend from each

end of the box to the most extreme values within 1.5 times the interquartile range from the ends of the box. The data with values beyond the ends of the whiskers, displayed as circles, are potential outliers. **d** Expression levels between H2A-K130ac and SREBF1 (nuclear) were significantly correlated in prostate tumors (n = 80 cores). For (**b** and **c**), p values were determined by Kruskal–Wallis test. For (**d**), the black line is the fitted linear line with 95% confidence interval bands shaded in pink. Source data are provided as a Source Data file.

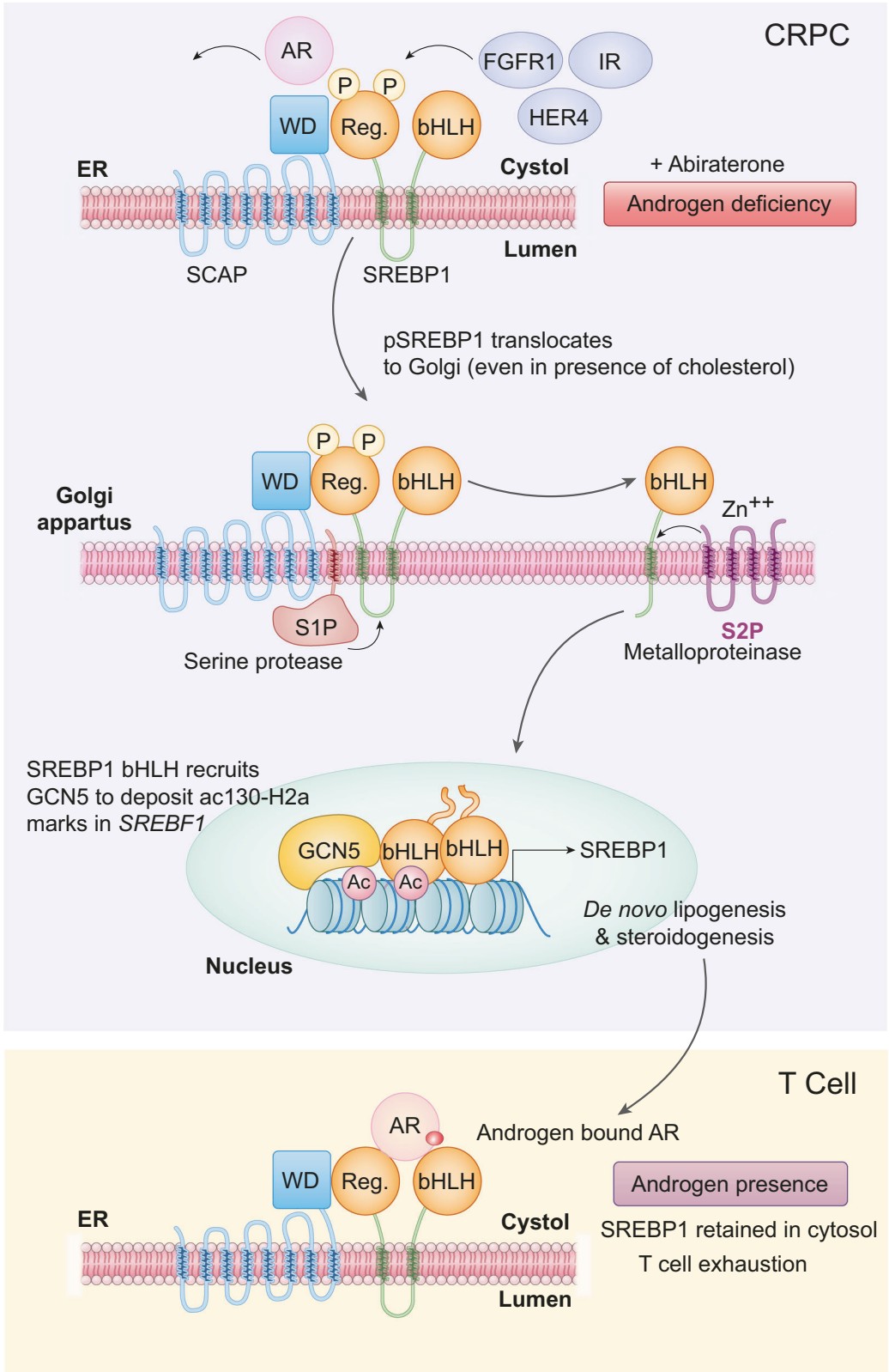

**Fig. 8 | pY-SREBF1/H2A-K130ac signaling senses androgen deficiency and regulates de novo androgen biosynthesis, a model.** Prostate cancer senses androgen by SREBF1 forming complex with androgen-bound-AR, preventing its nuclear localization. In androgen deficient environment, SREBF1 dissociated from androgen-unbound-AR, phosphorylated by tyrosine kinases, allowing its nuclear translocation, followed by recruitment of KAT2A to deposit acK130-H2A epigenetic marks in *SREBF1* exons. Epigenetic marking promoted *SREBF1* transcription, which in turn initiated cholesterologenic program, resulting in de novo androgen biosynthesis. Tumor derived androgen acted in paracrine manner on T cells, retaining SREBF1-androgen bound AR complex in cytosol, promoting T cell exhaustion.

an important molecular target to suppress tumor cell growth and activity[18,40]. Cholesterol lowering drugs such as statins or fatostatin have been suggested to help manage prostate cancer[41,42]. However, direct targeting of SREBFs therapeutically can potentially have deleterious effects in non-transformed cells, as engineered deletion at the *Srebf1* or *Srebf2* loci resulted in significant embryonic lethality[43,44]. In contrast, identification of pY-SREBF1/H2A-K130ac signaling nexus reveal an avenue to target cancer cell-specific steroidogenesis while sparing lipogenesis in normal cells. Indeed, a direct correlation of nuclear SREBF1 & H2A-K130ac with the disease progression to later stage in prostate cancer patients (Fig. 7), and recent report of elevated expression of KAT2A/GCN5 in patients with high-grade prostate cancer or recurrence following radical prostatectomy[45], endorse the importance of this signaling pathway. The combined inhibition of specific kinases, e.g., HER4 and KAT2A caused significant loss of tumor lipogenesis program leading to suppression of tumor-derived androgen (Fig. 6d, g), indicating its potential as a therapeutic strategy to overcome abiraterone-resistant CRPCs. Efficacy of abiraterone has spawned multiple studies to target other enzymes involved in androgen synthesis, including AKR1C3 (Aldo-Keto Reductase Family 1 Member C3), and STS (steroid sulfatase)[46]. However, similar to abiraterone, CRPCs are likely to acquire resistance to these compounds by employing pY-SREBF1/H2A-K130ac signaling and thus could be sensitized by inhibition of this nexus.

The CRPC patients exhibited gut microbiota that was capable of converting androgen precursors into active androgens, thus promoting endocrine resistance[47]. These data indicate the multi-factorial nature of abiraterone resistance in CRPC patients and would need specific combination of kinase inhibitors and anti-microbial agents to nullify the androgen level to get optimal clinical benefits.

Numerous studies indicate immunosuppressive effects of androgens[48,49]; however, the signaling pathway modulated in T cells by tumor-derived androgen remain poorly understood. These data demonstrate that pY-SREBF1/H2A-K130ac signaling-driven androgen synthesis in cancer cells when acted in paracrine manner, suppressed SREBF1 nuclear translocation in T cells, and caused increase in PD-1 and LAG3 exhaustion markers, leading to marked immunosuppression. Thus, the same signaling pathways operational in parallel in different cellular compartments seem to cause distinct consequences; cancer cells kick start androgen synthesis and continue to flourish; however, tumor-derived androgen promote T cells exhaustion.

A recent profiling study comparing benign and prostate cancer tissues of AA and EA ancestry reported race-specific alterations in certain lipid classes, especially TGs were found to be elevated in AA prostate cancer[39]. TGs are free fatty acid precursors shown to be an important energy source for proliferating prostate cancer cells[50,51]. Consistent with these data, our study reports high polyunsaturated TG levels in VCaP and C4-2B cells treated with abiraterone, suggesting that castration-resistant cells were able to promote proliferation in abiraterone-rich environment by augmenting the levels of polyunsaturated TGs. The concomitant decrease in levels of DGs in in C4-2B cells treated with abiraterone may be indicative of the increased propensity of proliferative tumors for lipogenesis i.e., quest for new energy sources (Supplementary Fig. 8). Increased levels of several phospholipid (PL) classes and PI were also observed in response to treatment with abiraterone (Supplementary Fig. 8–10); PLs are essential components of the plasma membrane and indicative of active PI3K/Akt signaling, similarly, PIs are important second messengers that participate in PI3K-Akt signaling, and also shown to be markers of aggressive proliferating tumors such as in AA men[52–55]. Overall, sustained de novo lipogenesis resulting in increased production of long-chain TGs, PLs and PI may contribute to the acquisition of resistance to abiraterone. Overall, lipid profile of androgen-deprived CRPCs and AA prostate cancer patients indicates that the pSREBF1/H2A-K130ac signaling could play a role in race-specific deregulation of lipid pathways.

In summary, evolutionary conservation of pSREBF1/H2A-K130ac signaling, and utility in both normal and cancer cell reveal vital importance of this signaling nexus. Consistent with that, KAT2A and tyrosine kinase inhibitor combination could have significant clinical impact for the patients with abiraterone-resistant prostate cancer.

## Methods

All the experiments for the study were performed following standards according to the protocol written and approved by the Institutional Biological & Chemical (IBC) Safety Committee, Washington University in St. Louis (IBC protocol no. 7100).

### Cell lines

VCaP and HEK293T cells were obtained from ATCC. C4-2B and LAPC4 cells were described earlier[20]. HEK293T and VCaP cells were cultured in DMEM, supplemented with 10% FBS. LAPC4 cells were cultured in IMDM, supplemented with 10% FBS. All cultures were maintained with 50 units/ml of penicillin/streptomycin (Invitrogen) and cultured in 5% CO2 incubator. All cultures were tested for mycoplasma contamination every 2 months using the PCR Mycoplasma Test Kit I/C (PromoKine). All cell lines tested negative for mycoplasma contamination. Identities of all cell lines were confirmed by Short Tandem Repeat (STR) Profiling.

### Mouse xenograft studies

All animal experiments were performed using the standards for humane care in accordance with the National Institutes of Health (NIH) Guide for the Care and Use of Laboratory Animals. Mice studies were performed according to IACUC protocols approved in writing by Washington University in St. Louis Department of Comparative Medicine (IACUC protocol no. 20180259). All mice were co-housed with 3–5 mice per cage and maintained in a controlled pathogen-free/germ-free environment with a temperature of 20-23 °C, 12/12 h light/dark cycle, 50–60% humidity, and food (PicoLab Rodent diet 5053) and water provided ad libitum. For all mice experiments, tumor volumes of 1500mm³ or end of treatment period, was considered as the humane end point. The maximum allowed weight loss was 10% of the body weight. The maximum limit for tumor burden and weight loss was not exceeded throughout the study.

Five- to six-week-old male SCID mice (CB17/Icr-*Prkdc^scid*/IcrIcoCrl Strain code: 236) were purchased from Charles River Laboratories, USA. $1.2 \times 10^6$ C4-2B cells infected with SREBF1 or Y673/951 A double mutant expressing constructs. Cells were suspended in 200 µl of PBS with 50% matrigel (BD Biosciences) and were implanted subcutaneously into the dorsal flank of SCID mice. Once the tumors reach approximately 100 mm³ in size (about 4 weeks), mice were gavaged orally with abiraterone (196 mg/Kg in cooking oil) or vehicle (10% DMSO in cooking oil), twice a week, for 4 weeks. Tumor volumes were measured twice weekly using calipers. At the end of the study, when the tumor volumes were approximately 1300 mm³, all mice were humanely euthanized by carbon dioxide inhalation, followed by cervical dislocation. Tumors were extracted and weighed.

Five- to six-week male C57BL/6 mice (Strain code: 027, Stock Number: C57BL/6-027) were purchased from Charles River Laboratories, USA. $1.2 \times 10^6$ TRAMP-C2 cells were suspended in 200 µl of PBS with 50% matrigel (BD Biosciences) and were implanted subcutaneously into the dorsal flank of C57BL/6 mice. Once the tumors reach approximately 100 mm³ in size (about 4 weeks), mice were gavaged orally with vehicle (10% DMSO) or abiraterone (96 mg/kg of body weight) or CPTH2+Afatinib (each 12 mg/kg of body weight) for five days a week, for 3 weeks. Tumor volumes were measured twice weekly using calipers. At the end of the study, when the tumor volumes were approximately 1400 mm³, all mice were humanely euthanized by carbon dioxide inhalation, followed by cervical dislocation. Tumors were extracted and weighed. The blood was collected and used for

testosterone and interferon-γ assessment. Tumors were used for RNA preparation followed by qRT-PCR.

Similarly, $1.5 \times 10^6$ C4-2B cells were suspended in 200 µl of PBS with 50% matrigel (BD Biosciences) and were implanted subcutaneously into the dorsal flank of male SCID mice. Once the tumors reach approximately 100 mm³ in size (about 4 weeks), mice were gavaged orally with vehicle (10% DMSO) or abiraterone (96 mg/kg of body weight) or CPTH2+Afatinib (each 12 mg/kg of body weight) for five days a week, for 3 weeks. Tumor volumes were measured twice weekly using calipers. At the end of the study, when the tumor volumes were approximately 700 mm³, all mice were humanely euthanized by carbon dioxide inhalation, followed by cervical dislocation. Tumors were extracted and weighed.

### Recombinant DNA transfection

HEK293T cells were transfected with control pcDNA3.1 (Vector) or SREBF1 or various mutants using X-tremeGENE HP DNA Transfection Reagent (Sigma). Transfected cells were grown in DMEM with 10% fetal bovine serum for 48 hr and/or treated with inhibitors/drugs and harvested for western blotting as per experimental conditions.

### Generation and affinity purification of the H2A-K130ac antibody

Two H2A peptides coupled to immunogenic carrier proteins were synthesized as shown below, and H2A-K130ac antibodies were custom synthesized by 21st Century Biochem, MA.

H2A Lys130-acetylated peptide: LPKKTESHHKAKGacK
H2A Lys130-acetylated peptide: LPKKTESHHKAKGK

Two rabbits were immunized twice with the acetyl peptide, and the sera from these rabbits was affinity purified. Two antigen-affinity columns were used to purify the acetyl-specific antibodies. The first column was the non-acetyl peptide affinity column. Antibodies recognizing the non-acetylated residues of the peptide bound to the column. The flow-through fraction was collected and then applied to the second column, the acetyl peptide column. Antibodies recognizing the acetyl residue bound to the column and were eluted as acetyl-specific antibodies.

### Immunoprecipitation and Western Blot Analysis

Treated cells were lysed by sonication in receptor lysis buffer (RLB) containing 20 mM HEPES (pH 7.5), 500 mM NaCl, 1% Triton X-100, 1 mM DTT, 10% glycerol, phosphatase inhibitors (50 mM NaF, 1 mM Na2VO4), and protease inhibitor mix (Roche). Lysates were quantitated and 20 to 100 mg of protein lysates were boiled in SDS sample buffer, size fractionated by SDS-PAGE, and transferred onto a PVDF membrane (GE Healthcare). After blocking in 3% bovine serum albumin (BSA), membranes were incubated with the following primary antibodies: anti-Myc (Cell Signaling Technology, Cat# 2276S, Clone 9B11, 1:1000), anti-Flag (Cell Signaling Technology, Cat# 14793S, Clone D6W5B, 1:1000), anti- phosphoTyrosine (pTyr) (Santa Cruz Biotechnology, Cat# SC-508, Clone PY20, 1:500), anti-SREBF1 (Santa Cruz Biotechnology, Cat# SC-365513, Clone A4, 1:1000), anti-HA (Cell Signaling Technology, Cat# 2367S, Clone 6E2, 1:1000), anti-Actin (Sigma, Cat#A2228, Clone AC-74, 1:9,000), anti-H2A (Cell Signaling Technology, Cat# 3636S, Clone L88A6, 1:1000), anti-KAT2A/GCN5 (Santa Cruz Biotechnology, Cat# SC365321, Clone A11, 1:1000), anti-AR (Santa Cruz Biotechnology, Cat#SC7305, Clone 441,1:1000). Following three washes in PBS-T, the blots were incubated with horseradish peroxidase-conjugated secondary antibody. The blots were washed thrice, and the signals visualized by enhanced chemiluminescence (ECL) system according to the manufacturer's instructions (Thermo Scientific) using Invitrogen™ iBright™ FL1000 imaging system.

For immunoprecipitation studies, cell lysed by sonication in RLB, the lysates were quantitated, and 0.5 to 1 mg of protein lysate was immunoprecipitated using 2 µg of anti-SREBF1, anti-pTyr, anti-FLAG, anti-HA, anti-Myc or anti-AR antibody coupled with protein A/G

sepharose (Santacruz) overnight, followed by washes with RLB and PBS buffers. The beads were boiled in a sample buffer and immuno-blotting was performed as described above. Densitometric analysis using ImageJ software (ImageJ, NIH, USA) was performed for each representative immunoblot image and actin normalized relative fold change intensity for each lane is incorporated, wherever required. All the images were compiled using Adobe Photoshop Version 24.x and Adobe Illustrator Version 26.4.

### Mass Spectrometry

HEK293T cells were transected with SREBF1 and FGFR-expressing constructs. Post 48 hr of transfection, cells were processed for LCMS/MS analysis. Samples were digested overnight with modified sequencing grade trypsin (Promega, Madison, WI), Glu-C (Worthington, Lakewood, NJ), or Arg-C (Roche,Switzerland). Phosphopeptides were enriched using Phospho Select IMAC resins (Sigma). A nanoflow ultra-high performance liquid chromatograph (RSLC, Dionex, Sunnyvale, CA) coupled to an electrospray bench top orbitrap mass spectrometer (Q-Exactive plus, Thermo, San Jose, CA) was used for tandem mass spectrometry peptide sequencing experiments. The sample was first loaded onto a pre-column (2 cm × 100 µm ID packed with C18 reversed phase resin, 5 µm, 100 Å) and washed for 8 min with aqueous 2% acetonitrile and 0.04% trifluoroacetic acid. The trapped peptides were eluted onto the analytical column, (C18, 75 µm ID × 50 cm, 2 µm, 100 Å, Dionex, Sunnyvale, CA). The 90-min gradient was programmed as: 95% solvent A (2% acetonitrile + 0.1% formic acid) for 8 min, solvent B (90% acetonitrile + 0.1% formic acid) from 5% to 38.5% in 60 min, then solvent B from 50% to 90% B in 7 min and held at 90% for 5 min, followed by solvent B from 90% to 5% in 1 min and re-equilibrate for 10 min. The flow rate on analytical column was 300 nl/min. Sixteen tandem mass spectra were collected in a data-dependent manner following each survey scan. Both MS and MS/MS scans were performed in Orbitrap to obtain accurate mass measurement using 60 second exclusion for previously sampled peptide peaks. Sequences were assigned using Sequest (Thermo) and Mascot (www.matrixscience.com) database searches against SwissProt protein entries of the appropriate species. Oxidized methionine, carbamidomethyl cysteine, and phosphorylated serine, threonine and tyrosine were selected as variable modifications, and as many as 3 missed cleavages were allowed. The precursor mass tolerance was 20 ppm and MS/MS mass tolerance was 0.05 Da. Assignments were manually verified by inspection of the tandem mass spectra and coalesced into Scaffold reports (www.proteomesoftware.com).

### Quantitative RT-PCR

All RT reactions were done at the same time so that the same reactions could be used for all gene studies. For the construction of standard curves, serial dilutions of pooled sample RNA were used (50, 10, 2, 0.4, 0.08, and 0.016 ng) per reverse transcriptase reaction. One "no RNA" control and one "no Reverse Transcriptase" control was included for the standard curve. Three reactions were performed for each sample: 10 ng, 0.8 ng, and a NoRT (10 ng) control. Real-time quantitative PCR analyses were performed using the ABI PRISM 7900HT Sequence Detection System (Applied Biosystems). All standards, the no template control (H₂O), the No RNA control, the no Reverse Transcriptase control, and the no amplification control (Bluescript plasmid) were tested in six wells per gene (2 wells/plate x 3 plates/gene). All samples were tested in triplicate wells each for the 10 ng and 0.8 ng concentrations. The no RT controls were tested in duplicate wells. PCR was carried out with SYBR Green PCR Master Mix (Applied Biosystems) using 2 µl of cDNA (or ChIP DNA) and the primers in a 20-µl final reaction mixture. After 2-min incubation at 50 °C, AmpliTaq Gold was activated by 10-min incubation at 95 °C, followed by 40 PCR cycles consisting of 15 s of denaturation at 95 °C and hybridization of primers for 1 min at 55 °C. Dissociation curves were generated for each plate to verify the integrity of the primers. Data were analyzed using StepOne

and StepOnePlus software version 2.3 and exported into an Excel spreadsheet. The actin or 18 s rRNA data were used for normalizing the gene values, i.e., ng gene/ng actin or 18 s rRNA per well. The primer sequences are shown in Supplementary Data 7.

## Chromatin immunoprecipitation (ChIP) and ChIP-Sequencing

After respective treatments as indicated, the cells were cross-linked in 0.75% formaldehyde solution for 10 min and Glycine was added to quench the formaldehyde and terminate the cross-linking reaction. Cell pellets were resuspended in RLB buffer and sonicated for 26 seconds (Branson sonicator). The soluble chromatins were incubated with indicated antibodies and Protein A/G magnetic beads (Bio-Rad) or anti-HA Affinity Gel (Millipore Sigma), for 3 h. The soluble chromatin was processed in the same way for IgG control and without immunoprecipitation for input DNA. The complexes were washed with RLB buffer three times, eluted with elution buffer. Cross-links were reversed by 2 h of incubation at 65 °C, followed by proteinase-K treatment (Active Motif). ChIP DNA was then purified using QIAGEN MinElute PCR Purification Kit. Ten nanograms of immunoprecipitated DNA were used to generate sequencing libraries using the Kapa Hyper Prep Kit (Roche Sequencing Solutions Inc., Pleasanton, CA). The size and quality of the library were evaluated using the Agilent BioAnalyzer (Agilent Technologies, Inc., Santa Clara, CA), and the library was quantitated with the Kapa Library Quantification Kit. Each enriched DNA library was then sequenced on an Illumina NextSeq 500 sequencer to generate 40-50 million 75-base paired-end reads (Illumina, Inc., San Diego, CA). The raw sequence data were aligned using BowTie 2[56], and the binding sites were identified using the MACS peak-finding software[57].

## Syngeneic tumor studies and flow cytometry

Male C57BL/6 mice (Strain code: 027, Stock Number: C57BL/6-027) were purchased from Charles River Laboratories, USA. $1.2 \times 10^6$ TRAMP-C2 cells were implanted 6- to 8-week-old C57BL/6 mice. When tumors were palpable, mice were gavaged orally with vehicle or abiraterone at (96 mg/kg), or CPTH2+Afatinib (each 12 mg/kg), 5 days a week, for 4 weeks. Tumor volumes were measured twice weekly using calipers. At the end of the study, all mice were humanely euthanized, tumors were extracted and weighed. The tumor-draining lymph nodes were collected and single-cell suspensions were made. For analysis of T-cell exhaustion, $1 \times 10^6$ cells were stained with Fixable Aqua Dead Cell stain (Thermo Fisher Scientific, Cat#L34957, 1:400) anti-CD3 PECy7 (BioLegend, Cat#100220, Clone 17A2, 1:400), anti-CD8 APC (BioLegend, Cat#553035, Clone 53-6.7, 1:400), anti-PD1 FITC (BioLegend, Cat#135213, Clone 29F.1A12, 1:400), anti-Lag3 Percpcy5.5 (BioLegend, Cat#125212, Clone C9B7W, 1:400) antibodies. Samples were analyzed using BD FACSCanto II (BD Biosciences) and post-acquisition analysis was done using FlowJo software (Tree Star Inc). Serum was collected and testosterone levels were assessed by steroid discovery assay (Eve Technologies, Calgary, AB).

## Intracellular calcium measurements

Splenocytes isolated from C57BL/6 mice were treated with vehicle or DHT for 1 h. Cells were loaded with Fluo-8 (Abcam) and incubated for 30 min at 37 °C. Intracellular calcium flux was measured using flow cytometry. For the analyzing the response to anti-CD3 antibody, dye-loaded cells were recorded for initial 30 s and then monitored after addition of anti-CD3 antibodies, followed by ionomycin addition at 300th sec.

## Hormone profiling

Serum samples from C57BL/6 mice with TRAMP-C2 tumors treated with vehicle or abiraterone or CPTH2+Afatinib were subjected to Mouse Steroid-Thyroid Hormone 6-Plex Discovery Assay (Eve Technologies, Calgary, AB).

## ELISA

Levels of IFN-γ in the blood serum of C57BL/6 mice with TRAMP-C2 tumors treated with vehicle or abiraterone or CPTH2+Afatinib was measured using mouse Interferon-α ELISA Kit (R&D Systems) according to manufacturer's protocol.

## Immunohistochemistry (IHC staining)

For assessment of SREBF1 and K130Ac-H2A expressions in the wildtype and Y673/951A xenografts, the tumors were excised, fixed with 10% formalin, paraffin sectioned and embedded. The sections were then rehydrataed and heated at 95 °C for 20 min in 10 mM Sodium Citrate buffer (pH 6.0) for antigen retrieval. The sections were then quenched with hydrogren peroxide briefly to block endogenous peroxidase activity, then permeabilized with 0.4% Triton X-100 in PBS. After blocking with normal horse serum (Vector Laboratories, Burlington, Ontario, Canada) for 30 min, the sections were then incubated with either mouse monoclonal SREBF1 antibody (Santa Cruz Biotechnology, Cat# SC-365513, Clone A4, 1:300 dilution) or rabbit polyclonal K130Ac-H2A antibody (1:300 dilution) at 4 °C overnight. The sections were incubated with biotinylated horse anti-mouse/rabbit IgG for 1 h each. The sections were developed with DAB peroxidase substrate and counterstained with hematoxylin (Vector Laboratories). All slides were then visualized under the microscope and images were captured using EVOS M5000 Imaging System.

## Tissue microarray (TMA) analysis

As a first step, an extensive validation of SREBF1 and K130Ac-H2A antibodies for the IHC staining in prostate tissues using various samples (positive and negative controls) was performed at different concentrations of antibodies. Negative controls were included by omitting SREBF1 and K130Ac-H2A antibodies during primary antibody incubation or HRP-labeled secondary antibody to negate background.

For the assessment of SREBF1 and K130Ac-H2A expression, immunohistochemistry was carried out on TMA (n = 80 cores). The tissue array slides (including positive and negative controls) were stained for the antibodies. The slides were dewaxed by heating at 65° Celsius for 60 min, washed two times, 15 min each, with xylene. Tissues were rehydrated by two series of 10 min washes in 100%, 95%, and 70% ethanol and distilled water. Antigen retrieval was performed by heating the samples at 95⁰C for 25 min in 10 mmol/L sodium citrate (pH 6.0). The slides were cooled in PBS for 30 min, with 10 min changes of PBS and permeabilized using 0.2% Triton-X100 in PBS for 10 min. Slides were washed with PBS + 0.2% Tween-20 for 10 min. After blocking with universal blocking serum (DAKO Diagnostic, Mississauga, Ontario, Canada) for 30 min, the samples were incubated with mouse monoclonal SREBF1 (1:300 dilution) and rabbit polyclonal K130Ac-H2A antibody (1:300 dilution) at 4⁰C overnight. The sections were incubated with biotin-labeled secondary and streptavidin-peroxidase for 30 min each (DAKO Diagnostic). The samples were developed with 3,39-diaminobenzidine substrate (Vector Laboratories, Burlington, Ontario, Canada) and counterstained with hematoxylin. Following standard procedures, the slides were dehydrated and sealed with cover slips. The SREBF1 and K130Ac-H2A staining were examined in a blinded fashion by pathologist (C.W.). The positive reactions were scored into four grades according to the intensity of staining: 0, 1+, 2+ and 3+.

## Global lipidomic analysis

Lipid extraction from cells, chromatographic separation technique, and analysis were performed[39]. Briefly, lipids were extracted by the Bligh-Dyer method[58] using water: methanol: dichloromethane (2:2:2) after spiking with internal standards pertaining to lipid classes. The organic layer was dried and suspended in Acetonitrile: water: isopropyl alcohol (10:5:85) with 10 mM Ammonium Acetate. Chromatographic separation was achieved by reverse-phase LC/MS and a Shimadzu CTO-20A Nexera X2 UHPLC system including a degasser, binary pump, and

a column oven[59]. A 1.8um particle 50 × 2.1mm Acquity HSS UPLC T3 column (Waters, Milford, MA, USA) was used to separate lipids. A TripleTOF 5600 with Turbo VTM ion source (AB Sciex, Concorn, ON, Canada) was used for data acquisition in positive and negative ionization modes. MS2 spectra were acquired using the data-dependent acquisition (DDA) function of the Analyst TF software (AB Sciex, Concord, ON, Canada) with dynamic exclusion for coverage depth. Missing values in the data were imputed using the K nearest-neighbor method (KNN). Data was log2 transformed and internal standard normalized (negative method-15:0-18:1(d7) PE (ISTD) and positive method −18:1(d7) Lyso PC (ISTD)), respectively.

For Fig. S8 (comparing Abiraterone-treated vs. untreated), differential lipids were determined by $t$ test ($p < 0.05$), followed by the Benjamini−Hochberg (BH) procedure for false discovery rate correction (FDR < 0.25). BH was used to control the probability of a Type I error rate due to the testing of multiple hypotheses. For Fig. 8 (comparing VCaP and C42B untreated vs Abiraterone-treated cells), one-way ANOVA method was performed ($p < 0.05$) followed by the Benjamini−Hochberg (BH) procedure for false discovery rate correction (FDR < 0.25). The normalized data for this study have been deposited to the Metabolomics Workbench (https://www.metabolomicsworkbench.org/) under the study 3296.

### Statistical analysis

All data are presented as mean ± SEM and all statistical parameters and analysis are mentioned in the figure legends, respectively. Data for all experiments were analyzed with GraphPad Prism 8.0 software. All statistical analyses were performed using Student's $t$ test or one-way ANOVA unless otherwise specified. $p$ values less than 0.05 were considered as statistically significant.

### Reporting summary

Further information on research design is available in the Nature Portfolio Reporting Summary linked to this article.

## Data availability

The normalized data for this study have been deposited to the Metabolomics Workbench (https://www.metabolomicsworkbench.org/) under the studyID 3296. The mass spec data is submitted in ProteomeXchange dataset PXD036894 ChIP-sequencing datasets generated in this study can be found at https://www.ncbi.nlm.nih.gov/geo/query/acc.cgi?acc=GSE206856 Source data are provided with this paper. The remaining data are available within the Article, Supplementary Information or Source Data file. Source data are provided with this paper.

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

## Acknowledgements

N.P.M. is a recipient of NIH/NCI grants (1R01CA208258, 5R01CA227025, and 1R01CA273054), Prostate Cancer Foundation (PCF) grant (17CHAL06) and Department of Defense award (PC200201). S.Y.R. Dent provided GCN5 and PCAF MEFs. The technical assistance of Petra Erdmann-Gilmore, Yiling Mi and Rose Connors is gratefully acknowledged. The Proteomics experiments were performed at the Washington University Proteomics Shared Resource (WU-PSR), which is supported by the WU Institute of Clinical and Translational Sciences (NCATS UL1 TR000448), the Mass Spectrometry Research Resource (NIGMS P41 GM103422) and the Siteman Comprehensive Cancer Center Support Grant (NCI P30 CA091842). This work has been supported in part by the Proteomics and Metabolomics Core Facility at the Moffitt Cancer Center, an NCI designated Comprehensive Cancer Center (P30-CA076292). The technical assistance of Vasanta Putluri and Danthasinghe Waduge Badrajee Piyarathna for global lipidomics profiling is gratefully acknowledged. This project was supported by CPRIT Proteomics and Metabolomics Core Facility to N.P. (RP210227), NIH (P30 CA125123), and Dan L. Duncan Cancer Center.

## Author contributions

T.N., D.S., S.C., and A.W. performed the experiments; T.L. analyzed ChIP-seq data; C.W. performed pathologist's analysis; J.L. performed statistical analysis; J.K. and B.F. performed mass spectrometry studies; N.P. and A.S. performed global lipidomics analysis; F.Y.F. and K.M. analyzed the data, N.P.M. conceived the study, designed the experiments, acquired funding and wrote the manuscript. All authors read the manuscript.

## Competing interests

A patent "Inhibitors of ACK1/TNK2 Tyrosine Kinase" (patent no. 9,850,216; 10,017,478 and 10,336,734) covers (R)-9b compound. NPM

and KM are named as inventors. These patents have been licensed by TechnoGenesys Inc. KM and NPM are co-founders of TechnoGenesys Inc., own stocks, and serve as consultants for TechnoGenesys Inc. The other authors declare no potential conflicts of interest..
