## [Peer Review File · Nature Communications]

Histone H2A Lys130 Acetylation Epigenetically Regulates Androgen Production in Prostate CancerReviewers' Comments:

Reviewer #1:

Remarks to the Author:

In this complex yet well-written and fascinating manuscript, Nguyen et al uncover mechanisms regulating prostate cancer (PC) cells androgen sensing that are possibly activated when patients undergo androgen deprivation therapy (ADT), and that could be implicated in the emergence of castration resistant prostate cancer (CRPC). The authors discover a new histone modification, H2AK130ac, that increases presumably with shortage of androgens in PC cells. The deposition of the mark seems selective for genomic region in the SREBF1 gene locus and this leads to its transcriptional activation and phosphorylation of the resulting protein specifically in two sites identified by the authors: Y673/951F. The authors show that SREBF1 interacts with AR in androgen dependent manner and postulate that absence of androgens reverts this interaction and exposes these phosphorylation sites that are responsible for nuclear localisation of SREBF1. They also show that the sites are phosphorylated in presence of FGF, putatively by FGFR, and in presence of Insulin, putatively by HER4. In turn, the authors describe a feed forward loop in which nuclear, phosphorylated SREBF1, recruits GCN5, which is the identified HAT, able to deposit H2A-130Ac. Ultimately is this feedforward loop that supports the activation of steroidogenesis in castrate conditions.

Furthermore, as supporting evidence, the authors show that:

- Reversal of H2AK130ac or pY-SREBP1 in C42B xenografts sensitised to androgen synthesis inhibitor, Abiraterone
- IHC staining of Nuclear SREBP1 and H2A-K130ac levels were significantly increased and directly correlated with late stage prostate cancer patients;
- Global lipidomics signature resembles PCs from African-American men who are notoriously at higher risk of dying for this disease.

This massive amount of work is important because it raises new intriguing hypotheses and calls for verifying testable treatments combinations for particular classes of PC patients.

I have some major comments:

1. Fig. 1E and Fig S1E: It is not clear for how long the cells were serum starved before assessment of levels of H2A-K130Ac. Did the androgen or serum starvation impact on the H2A-K130Ac levels? Did androgen starvation impact the SREBP1-AR interaction? Fig. 6A-C do not show this directly.
2. Fig. S1F in HEK cells: the reason why the authors focus the rest of the paper on KAT2A/GCN5 and e.g., not PCAF is not clear. The consequence of exposure of HAT-overexpressing prostate cancer lines on the H2A-K130Ac could be tested in presence and absence of Abiraterone as in Fig. 1E and S1E. Also, IP with SREBP1 could equally be performed with the rest of the HATs as in Fig. 4G. Is maybe the selective interaction with SREBP1 that determines its genomic targets, and possibly explains the focus of the paper?
3. The H2A-K130Ac could be better presented, especially because it is not clear whether the target of increased H2A-K130Ac is indeed primarily the transcription of SREBF1.
 - 3a. How many sites were significantly differentially Acetylated upon Romidepsin treatment? Were new sites created?
 - 3b. In Fig 2A please show also the DMSO track.
 - 3c. Are the selected sites for validation resulting from the enrichment over the DMSO control?
 - 3d. Fig. S2D and 2C controls are missing (please show enrichment of IgG at the same sites).
 - 3e. In Suppl. Table 2 the peaks are named "vcap_romi_minus_vcap_input_peak_474" which makes me think that the sites were selected merely on their enrichment above input, and not because they were enriched due to the treatment with Romidepsin. Can you rank the sites according to their enrichment vs Romidepsin?
 - 3f. Figure S2C: which intervals were selected for motif analysis? A comparison between the motifs enriching in the DMSO and in the Romidepsin ChIPped regions should be shown.
4. Fig. 2D: Is SREBF1 transcription and transcription of its targets consistently activated by Abi also in VCaP cells?

5. Fig 2G: validation of the constructs should be shown in Supplementary. Are the treatments with Romi or Abi affecting the expression of SREBF1 target genes differently in the mutant bearing cells?
6. Fig. 4H-I: is the interaction of GCN5 and SREBP1 DNA-dependent? Should KAT2A read GCN5 consistently throughout the manuscript?
7. Fig. 5. C42B is a metastatic model. Where metastases investigated in the xenografts experiment? Where the mice intact, castrated, or treated with agents to reduce testosterone production?
8. Fig. 6E: Although I understand that n=3, the volumes of the tumors should be presented e.g. similar to Fig. 5B.
9. Fig. 7A. Some magnifications that allow to clearly appreciate the nuclear should be added.
10. The discussion regarding the immunology related part and the lipidomics data reads a little too speculative and could be shortened.

I also have some minor comments:

1. Fig. S4E regards LAPC4 and should be corrected in the text.
2. The work by Pernigoni et al., (<https://www.science.org/doi/10.1126/science.abf8403>) is worth to be mentioned in the introduction around ref. 12 or in discussion.
3. Since the authors do not distinguish the SREBP1 protein products from the SREBF1 maybe the official name could be used for the protein too.
4. Considering the results in this cited paper: <https://www.nature.com/articles/s41586-022-04522-6> in which inhibition of AR activity in CD8 T cells was shown to prevent T cell exhaustion and improve responsiveness to PD-1 targeted therapy via increased IFN γ expression, do the author could see any IFN γ expression modulation in their experiments with the TRAMP model in the CP+Afa conditions?

Reviewer #2:

Remarks to the Author:

Nguyen T et al. in this interesting study reported a mechanistic pathway that provides prostate cancer cells the ability to sense the lack of androgen to support androgen de novo synthesis and consequently castration resistance and tumour progression. The authors use a plethora of molecular pharmacology techniques, in vitro and in vivo studies, and finally clinical data. Their method sound and the results support the conclusions and claims. In addition, author provided a link between androgen-SREBP1 and T cell exhaustion, which will open future research to explore immunotherapy targeting via SREBP1 for prostate cancer treatment.

The article topic is very interesting, novel and would have huge impact on prostate cancer research.

Major comments:

1. Androgen deprivation was tested only using 1 model, abiraterone. Another model such as culturing the cells in Charcoal Stripped Fetal Bovine Serum to validate the effects and preclude an abiraterone-selective effect hypothesis.
2. While the authors tested the therapeutic efficacy of targeting the identified pathway using in vitro and in vivo models, utilising a human-derived tissue (explants, organoids, or xenografts) is warranted.

Minor comments:

1. Fig 3F, correct the IP, WB labels.
2. In vivo C4-2B studies, the tumour in control group reached 1300 mm³, this could be a huge burden on the mice. Is this size acceptable in the animal ethics approval?
3. Fig 5G, depicting the results in a graph would be easier to be interpreted.

Reviewer #3:

Remarks to the Author:

In this work, the authors discovered a heretofore uncovered epigenetic mark, K130Ac on H2A, following dual-phosphorylation on SREBP1, and demonstrated this mark is key to transactivate de novo androgen synthesis to overcome the pharmacological inhibition of androgen synthesis.

Although this discovery may be functionally important, the omics analysis is flawed.

1. The dual-phosphorylated SREBP is questionable. The pY951-SREBP spectrum is of very low quality and the peptide is poorly covered by available MS/MS ions. I wouldn't assign Fig. S3B as a match.

2. No valuable details can be found for phosphorylated peptides enrichment. MS data acquisition lacks necessary details for both of the proteomics and lipidomic sections.

3. Again, the assignment of the very important epigenetic mark, K130Ac on H2A, lacks fidelity. The representative MS/MS spectrum the authors chose to show in Fig. 1A carries the acetylation modification on the C-terminal K. However, it is a commonsense that acetylated lysines cannot be cleaved by trypsin in most cases. Another reason is that this selected peptides carries multiple miss-cleavages, such as the two consecutive K in the middle of the sequence.

Point-by-point responses to the Reviewer's comments

Reviewer #1, expertise in androgen receptor signalling, treatment resistance, epigenomics, prostate cancer (Remarks to the Author):

In this complex yet well-written and fascinating manuscript, Nguyen et al uncover mechanisms regulating prostate cancer (PC) cells androgen sensing that are possibly activated when patients undergo androgen deprivation therapy (ADT), and that could be implicated in the emergence of castration resistant prostate cancer (CRPC). The authors discover a new histone modification, H2AK130ac, that increases presumably with shortage of androgens in PC cells. The deposition of the mark seems selective for genomic region in the SREBF1 gene locus and this leads to its transcriptional activation and phosphorylation of the resulting protein specifically in two sites identified by the authors: Y673/951F. The authors show that SREBF1 interacts with AR in androgen dependent manner and postulate that absence of androgens reverts this interaction and exposes these phosphorylation sites that are responsible for nuclear localisation of SREBF1. They also show that the sites are phosphorylated in presence of FGF, putatively by FGFR, and in presence of Insulin, putatively by HER4. In turn, the authors describe a feed forward loop in which nuclear, phosphorylated SREBF1, recruits GCN5, which is the identified HAT, able to deposit H2A-130Ac. Ultimately is this feedforward loop that supports the activation of steroidogenesis in castrate conditions. Furthermore, as supporting evidence, the authors show that:

- Reversal of H2AK130ac or pY-SREBP1 in C42B xenografts sensitised to androgen synthesis inhibitor, Abiraterone
- IHC staining of Nuclear SREBP1 and H2A-K130ac levels were significantly increased and directly correlated with late stage prostate cancer patients
- Global lipidomics signature resembles PCs from African-American men who are notoriously at higher risk of dying for this disease.

This massive amount of work is important because it raises new intriguing hypotheses and calls for verifying testable treatments combinations for particular classes of PC patients.

Response: Thank you so much for such a careful review of our manuscript.

I have some major comments:

1. Fig. 1E and Fig S1E: It is not clear for how long the cells were serum starved before assessment of levels of H2A-K130Ac. Did the androgen or serum starvation impact on the H2A-K130Ac levels? Did androgen starvation impact the SREBP1-AR interaction?

Response: Serum starvation can lead to androgen deprivation; accordingly, similar to Abiraterone treatment, the serum starvation too caused increase in H2A-K130Ac, this data is now enclosed (**Supplementary Fig S1f**)

Considering the possibility that presence of FBS could partly nullify the effect of Abiraterone, in **Fig. 1e** and **Fig S1e**, we resuspended cells in –FBS media, and then added Abiraterone. We have made edits in the figure legends to reflect these experimental conditions.

Expectedly, serum starvation compromised SREBP1-AR interaction (**Supplementary Fig S1f**).

2. Fig. S1F in HEK cells: the reason why the authors focus the rest of the paper on KAT2A/GCN5 and e.g., not PCAF is not clear. The consequence of exposure of HAT-overexpressing prostate

cancer lines on the H2A-K130Ac could be tested in presence and absence of Abiraterone as in Fig. 1E and S1E. Also, IP with SREBP1 could equally be performed with the rest of the HATs as in Fig. 4G. Is maybe the selective interaction with SREBP1 that determines its genomic targets, and possibly explains the focus of the paper?

Response: Both KAT2A and PCAF seem to interact with SREBP1 in phosphorylation dependent manner (**Figure 4g, Supplementary Fig S5a and c**). However, upon Abiraterone treatment, H2A-K130ac levels were increased in presence of GCN5, while, the increase was not seen in PCAF expression (**Supplementary Fig S5d**). This does not completely rule out the potential role of PCAF in deposition of H2A-K130ac marks in abiraterone dependent manner, however, we focused on KAT2A for this manuscript. We believe, multiple HATs could be involved H2A-K130ac marks deposition depending on cell type or disease stage.

3. The H2A-K130Ac could be better presented, especially because it is not clear whether the target of increased H2A-K130Ac is indeed primarily the transcription of SREBF1. 3a. How many sites were significantly differentially Acetylated upon Romidepsin treatment? Were new sites created?

Response: We have over 1400 peaks in Romidepsin treated samples, 155 peaks in DMSO sample and 28 overlapping peaks between these two samples. All these peak data is shown in **Supplementary Table S1**, as well as in form of a Venn diagram (**Supplementary Fig S2c**).

3b. In Fig 2A please show also the DMSO track.

Response: It is now included.

3c. Are the selected sites for validation resulting from the enrichment over the DMSO control?

Response: Yes

3d. Fig. S2E controls are missing (please show enrichment of IgG at the same sites).

Response: IgG data is added

3e. Figure S2C: which intervals were selected for motif analysis?

Response: We used default 5% intervals for motif finding.

4. Fig. 2D: Is SREBF1 transcription and transcription of its targets consistently activated by Abi also in VCaP cells?

Response: Yes, the data in VCaP cells is now added, please see **Figure 2d** and **Supplementary Figure S2g**.

5. Fig 2G: validation of the constructs should be shown in Supplementary. Are the treatments with Romi or Abi affecting the expression of SREBF1 target genes differently in the mutant bearing cells?

Response: Western with expression is shown in **Supplementary Figure S2h**.

The treatments with Romi or Abi did not affect the expression of SREBF1 target genes in the mutant bearing cells (**Supplementary Figure S2i-k**).

6. Fig. 4H-I: is the interaction of GCN5 and SREBP1 DNA-dependent? Should KAT2A read GCN5 consistently throughout the manuscript?

Response: We purified FLAG-tagged KAT2A and MYC-tagged N-SREBP1 from HEK293 transfected cells and assessed their interaction *ex vivo* (**Supplementary Figure S5b**). It appears that KAT2A could interact with N-SREBP1 in DNA independent manner.

7. Fig. 5. C42B is a metastatic model. Where metastases investigated in the xenografts experiment? Where the mice intact, castrated, or treated with agents to reduce testosterone production?

Response: Yes, the C4-2B cells have ability to form metastatic tumors and we have published that in past (*Cancer Cell*, 2017). These sets of experiments were not done with the intent to study metastasis.

We did not have to castrate the mice, as the abiraterone is supposed to reduce testosterone production (chemical castration). Consistently, we noticed that C4-2B cells, being CRPC type, were able to form tumors even in presence of Abi, however, the mutant SREBF1 construct expressing C4-2B significantly compromised their tumor formation ability (**Figure 5b-d**).

8. Fig. 6E: Although I understand that n=3, the volumes of the tumors should be presented e.g. similar to Fig. 5B.

Response: OK. We have shown the tumor volume data in **Supplementary Figure S6b**.

9. Fig. 7A. Some magnifications that allow to clearly appreciate the nuclear should be added.

Response: IHC magnifications are now added.

10. The discussion regarding the immunology related part and the lipidomics data reads a little too speculative and could be shortened.

Response: We have shortened immunology and the lipidomics data discussion.

I also have some minor comments:

1. Fig. S4E regards LAPC4 and should be corrected in the text.

Response: Thank you for pointing it, we have corrected that.

2. The work by Pernigoni et al., (<https://www.science.org/doi/10.1126/science.abf8403>) is worth to be mentioned in the introduction around ref. 12 or in discussion.

Response: We have added this reference in the discussion section.

3. Since the authors do not distinguish the SREBP1 protein products from the SREBF1 maybe the official name could be used for the protein too.

Response: Sure, we have changed SREBP1 to SREBF1.

4. Considering the results in this cited paper: <https://www.nature.com/articles/s41586-022-04522-6> in which inhibition of AR activity in CD8 T cells was shown to prevent T cell exhaustion and improve responsiveness to PD-1 targeted therapy via increased IFN γ expression, do the author could see any IFN γ expression modulation in their experiments with the TRAMP model in the CP+Afa conditions?

Response: Blood serum IFN gamma levels by ELISA is shown in **Figure 6k**.

Reviewer #2, expertise in lipid metabolism and prostate cancer (Remarks to the Author):

Nguyen T et al. in this interesting study reported a mechanistic pathway that provides prostate cancer cells the ability to sense the lack of androgen to support androgen *de novo* synthesis and consequently castration resistance and tumour progression. The authors use a plethora of molecular pharmacology techniques, *in vitro* and *in vivo* studies, and finally clinical data. Their method sound and the results support the conclusions and claims. In addition, author provided a link between androgen-SREBP1 and T cell exhaustion, which will open future research to explore immunotherapy targeting via SREBP1 for prostate cancer treatment. The article topic is very interesting, novel and would have huge impact on prostate cancer research.

Response: Thank you so much for reviewing our manuscript.

Major comments:

1. Androgen deprivation was tested only using 1 model, abiraterone. Another model such as culturing the cells in Charcoal Stripped Fetal Bovine Serum to validate the effects and preclude an abiraterone- selective effect hypothesis.

Response: We have included data for the second model, TRAMP-C2 cells. Here, castrated (androgen deprived) C57BL/6 mice were injected with TRAMP-C2 cells, and were sensitized by CPTH2+Afinib inhibitor combination (**Figure 5g**)

2. While the authors tested the therapeutic efficacy of targeting the identified pathway using *in vitro* and *in vivo* models, utilising a human-derived tissue (explants, organoids, or xenografts) is warranted.

Response: We have performed the experiment with human CRPC model. C4-2B xenografts were insensitive to Abiraterone treatment, however, their growth was compromised upon CPTH2+Afinib inhibitor combination treatment (**Figure 5h and i**)

Minor comments:

1. Fig 3F, correct the IP, WB labels.

Response: Thanks for pointing out, we have corrected it.

2. *In vivo* C4-2B studies, the tumour in control group reached 1300 mm³, this could be a huge burden on the mice. Is this size acceptable in the animal ethics approval?

Response: Yes, 1300 mm³ tumor weighs less than 1.5 gm. By our IACUC protocol, we can have tumors grown in mice up to 2.5 gm, which is about 10% of the weight of mice.

Reviewer #3, expertise in mass-spec based proteomics and metabolomics (Remarks to the Author):

In this work, the authors discovered a heretofore uncovered epigenetic mark, K130Ac on H2A, following dual-phosphorylation on SREBP1, and demonstrated this mark is key to transactivate *de novo* androgen synthesis to overcome the pharmacological inhibition of androgen synthesis. Although this discovery may be functionally important, the omics analysis is flawed.

1. The dual-phosphorylated SREBP is questionable. The pY951-SREBP spectrum is of very low quality and the peptide is poorly covered by available MS/MS ions. I wouldn't assign Fig. S3B as a match.

Response: Thank you for the comments. We have re-performed the mass spectrometry studies, the new pY951-SREBP spectrum is now included, please refer to **Supplementary Figure S3b**.

2. No valuable details can be found for phosphorylated peptides enrichment. MS data acquisition lacks necessary details for both of the proteomics and lipidomic sections.

Response: The entire mass spec data is now deposited in the database. Please follow as described below:

For reviewer to view Massive dataset:

- 1 Go to the “Massive.ucsd.edu” website
- 2 The reviewer should sign in to Massive using “MSV000090361_reviewer” as the User name, and the password “Mahajan_004_006” to access the dataset for reviewing.

Please Login to Use Workflows

MassIVE Repository Statistics			
Public Datasets:	12,373	Proteins:	20,363
Number of Files:	7,183,800	Peptides:	7,999,432
Total Size:	423.87 TB	Peptide Variants:	21,857,765
Spectra:	5,220,586,937	PSMs:	982,559,448
Dataset Subscriptions:	8,439	Modifications:	1,282

Full member of the **Proteome Xchange** consortium

Search Dataset Identifiers or Metadata: Search
Search Universal Spectrum Identifier (USI): Search
(example USI)

- 3 The Massive title page will open. Enter the name of the dataset “MSV000090361” and hit enter to open the web page for your dataset.
- 4 For the reviewer to view the data, they should FTP to <ftp://MSV000089589@massive.ucsd.edu> to download the data.

The lipidomics data for this study have been deposited to the Metabolomics Workbench (<https://www.metabolomicsworkbench.org/>) under the studyID 3296.

3. Again, the assignment of the very important epigenetic mark, K130Ac on H2A, lacks fidelity. The representative MS/MS spectrum the authors chose to show in Fig. 1A carries the acetylation modification on the C-terminal K. However, it is a commonsense that acetylated lysines cannot be cleaved by trypsin in most cases. Another reason is that this selected peptides carries multiple miss-cleavages, such as the two consecutive K in the middle of the sequence.

Response: Under most normal circumstances, this would be correct. In tryptic peptides internal within the substrate protein, the C-terminal lysine cannot be acetylated, because trypsin has to recognize the positive charge of lysine or arginine in the binding pocket prior to cleavage. However, K130 is the C-terminus of the whole H2A Type 1-C protein (<https://www.uniprot.org/uniprotkb/Q93077/entry>). Therefore, the trypsin cleavage rules do not

apply to that residue, and the C-terminus of the protein can be acetylated. This tryptic peptide is cleaved at R100 and not further cleaved at the lysines that remain in the sequence. The detection of the y₂ ion in the spectrum indicates that a fragment corresponding to G(Ac)K was detected, which confirms that the C-terminus is acetylated.

The concern about the number of missed cleavages also has merit, because 4 missed cleavages to make this assignment is quite high. However, the data in the tandem spectrum here are consistent with the sequence; for example, the overall sequence coverage is quite good (please see top of the Figure 1a) and individual clues like the dominant signals for y₁₃ and y₁₃₋₂₊ ions corresponding to fragments caused by proline-directed cleavage, PKKTESHHKAKG(AcK), are also consistent with the assignment. The missed cleavages at the terminus of the protein are also consistent with trypsin, which usually performs better for cleaving proteins than it does for sequentially cleavage adjacent or nearby tryptic sites.

Reviewers' Comments:

Reviewer #1:

Remarks to the Author:

The authors have responded to all my comments appropriately and the reviewer is satisfied with the revised version.

I have two minor clarifications:

1) In response to my comment number 2 the authors write: "This does not completely rule out the potential role of PCAF in deposition of H2A-K130ac marks in abiraterone dependent manner, however, we focused on KAT2A for this manuscript. We believe, multiple HATs could be involved H2A-K130ac marks deposition depending on cell type or disease stage."

"We believe, multiple HATs could be involved H2A-K130ac marks deposition depending on cell type or disease stage." could be emphasised in the text or in the discussion.

2) In my comment 3e. "Figure S2C: which intervals were selected for motif analysis?" I meant to ask which genomic intervals were used for the motif analysis. This could be specified in the M&M section and or in the figure legend.

Congratulations for your impactful study.

Reviewer #2:

Remarks to the Author:

The authors has answered all my comments.

Reviewer #4:

None

Point-by-point response

Reviewer #1 (Remarks to the Author):

The authors have responded to all my comments appropriately and the reviewer is satisfied with the revised version.

I have two minor clarifications:

1) In response to my comment number 2 the authors write: "This does not completely rule out the potential role of PCAF in deposition of H2A-K130ac marks in abiraterone dependent manner, however, we focused on KAT2A for this manuscript. We believe, multiple HATs could be involved H2A-K130ac marks deposition depending on cell type or disease stage."
"We believe, multiple HATs could be involved H2A-K130ac marks deposition depending on cell type or disease stage." could be emphasised in the text or in the discussion.

Response: Done. The places described in manuscript is now highlighted (please see pages 6 and 10).

2) In my comment 3e. "Figure S2C: which intervals where selected for motif analysis?" I meant to ask which genomic intervals were used for the motif analysis. This could be specified in the M&M section and or in the figure legend.

Response: We used default 5% intervals for motif finding.

Congratulations for your impactful study.

Response: Thank you so much.

Reviewer #2 (Remarks to the Author):

The authors has answered all my comments.

Response: Thank you so much.